# Rethinking RoPE Scaling in Quantized LLM: Theory, Outlier, and Channel-Band Analysis with Weight Rescaling

## Abstract

Extending the context window support of large language models (LLMs) is crucial for tasks with long-distance dependencies. RoPE-based interpolation and extrapolation methods, such as linear scaling and frequency-aware schemes, enable longer input length support without retraining, while post-training quantization (PTQ) makes deployment practical. However, we show that *combining* RoPE position interpolation (PI) with PTQ degrades accuracy due to coupled effects including long-context aliasing, dynamic-range dilation, anisotropy from axis-aligned quantizers vs. rotated RoPE pairs, and outlier shifting that produces position-dependent logit noise. We provide, to the best of our knowledge, the first systematic analysis of the PI+PTQ approach and introduce two practical diagnostics: *interpolation pressure* (per-band sensitivity to phase scaling) and *tail-inflation ratios* (outlier shift from short to long contexts). Following the analysis results, we propose **Q-ROAR** (Quantization, RoPE-interpolation, and Outlier Aware Rescaling), a *weight-only*, interpolation-aware stabilization of PI for quantized LLMs. Q-ROAR groups RoPE dimensions into a small number of frequency bands and performs a lightweight search over per-band scales for *Key* and *Query* weights (with an optional symmetric variant to preserve logit scale). The search is guided by our diagnostics and uses a tiny long-context development dataset, requiring no fine-tuning to the model, no architecture or kernel changes, and no additional deployment overhead. Empirically, Q-ROAR reduces the model's perplexity on long-context workloads by more than 14%, while preserving short-context performance, inference throughput, and compatibility with existing LLM system stacks.

## 1 Introduction

Large language models (LLMs) such as GPT-3 (Brown et al., 2020), LLaMA (Touvron et al., 2023), and DeepSeek-R1 (Guo et al., 2025) have advanced translation, coding, question answering, and dialogue. Yet their utility is often capped by the pretrained context window. Extending context is crucial for long-form summarization (Liu et al., 2023), code completion, retrieval-augmented generation (RAG) (Lewis et al., 2021), chain-of-thought (Wei et al., 2023), and agentic pipelines, which rely on long-range (long-distance) dependencies and rich cross-document context.

A popular path to longer contexts is to modify positional encoding at inference. RoPE-based scaling methods including linear interpolation (Chen et al., 2023a), frequency-aware YaRN (Peng et al., 2023), and evolutionary LongRoPE (Ding et al., 2024). They warp per-dimension phases so the model can read beyond its original window without retraining. Conceptually, many such methods fit a unified view in which positions are warped and per-dimension phase growth is rescaled. This preserves the rotation's orthogonality while changing how fast phases accumulate.

Meanwhile, practical deployment increasingly depends on **post-training quantization (PTQ)** such as GPTQ (Frantar et al., 2022), SmoothQuant (Xiao et al., 2023), and AWQ (Lin et al., 2024) to reduce memory footprint and latency across servers and edge devices. PTQ operates with diverse precision settings (weight-only quantization or weight+activation quantization). A well-known challenge for quantized model is activation outliers: rare, high-magnitude coordinates inflate quantization clipping ranges and waste effective bits, degrading accuracy (Liu et al., 2024).

We observe that simply applying RoPE **position interpolation (PI)** to PTQ models degrades accuracy both inside and especially beyond the original window. Our analysis attributes this to *coupled* mechanisms: (i) *long-context aliasing* from rapidly wrapping phases; (ii) *dynamic-range dilation* as pre-activations shift under PI; (iii) *anisotropy* because axis-aligned quantizers see RoPE-rotated pairs at changing angles; and (iv) *outlier shifting/amplification* as tails move and concentrate under new phase trajectories. We formalize a sensitivity indicator, *interpolation pressure*, which rises for higher-frequency bands, and show how PI and quantization errors interact in attention logits through position-dependent perturbations.

We propose **Q-ROAR** (Quantization, RoPE-interpolation, and Outlier Aware Rescaling) to mitigate the challenges. Q-ROAR groups RoPE dimensions into a small number of frequency *bands* and searches for lightweight, per-band scales for $(W_Q, W_K)$ that minimize perplexity on a tiny long-context development dataset. Two simple diagnostics guide a safe, light search: (i) *interpolation pressure* to avoid over-perturbing high-frequency bands; and (ii) *tail-inflation ratios* that summarize PI-induced outlier shift from short to long contexts. We adopt a symmetric scaling option ($W_Q \leftarrow g_b W_Q$, $W_K \leftarrow g_b^{-1} W_K$) to keep logit magnitudes stable. The method is a drop-in, quantizer- and backend-agnostic adjustment that requires no fine-tuning, no architecture changes, and no additional runtime overhead. Our contribution can be summarized as follows.

- **Analysis of the coupled PI+PTQ approach.** We provide a unified analysis which explains why PI harms quantized models: aliasing, dynamic-range dilation, anisotropy, and outlier shift reinforce one another and manifest as position-dependent logit noise.

- **Actionable diagnostics.** We introduce *interpolation pressure* (per-band PI sensitivity) and *tail-inflation* (outlier shift from short to long contexts) as practical signals to guide robust, frequency-selective interventions.

- **Weight-only, band-limited rescaling.** We present Q-ROAR, which searches over a tiny, log-spaced grid of per-band *weight* scales for $(W_Q, W_K)$, with symmetric scaling to preserve logit scale and safe bounds to avoid clipping or underflow in quantization.

- **Portability and improved performance.** Q-ROAR integrates with common PTQ baselines and improves long-context performance across benchmarks, while maintaining compatibility with existing computing kernels and LLM serving system stacks. We report consistent performance improvements ($> 14\%$) on extended-context evaluations with no inference overhead and no retraining.

## 2 BACKGROUND AND PRELIMINARY STUDY

### 2.1 RoPE BASICS

Unlike recurrent neural networks, transformer models do not have natural position information and rely on added positional encoding. The original transformer model (Vaswani et al., 2017) utilizes sinusoidal absolute positional encoding, while it claims similar results with learnable encoding. Rotary Position Embedding (RoPE) (Su et al., 2024) on the other hand, as a method that consider both absolute and relative positional information, has been widely adopted in transformer-based LLMs. RoPE injects position information by applying a position-dependent *rotation* to each adjacent 2D coordinate pair of the $d$-dimensional vector (hidden dimension), yielding attention that depends on *relative* positions while preserving norms. Let $x \in \mathbb{R}^d$ with even $d$ (if $d$ is odd, a trailing zero can be appended in practice). Define a frequency vector $\boldsymbol{\theta} \in \mathbb{R}_{>0}^{d/2}$ with components $\theta_i = b^{-2i/d}$ for $i = 0, \ldots, \frac{d}{2} - 1$ and base $b{=}10{,}000$ unless stated otherwise. For a position index $p \in \mathbb{N}$, the RoPE transform is the linear map

$$\text{RoPE}(x, p) = \mathbf{R}(p) x, \qquad \mathbf{R}(p) = \text{diag}\Big(\mathbf{R}_2(p\theta_0), \ldots, \mathbf{R}_2(p\theta_{\frac{d}{2}-1})\Big), \tag{1}$$

where each $2{\times}2$ block is the planar rotation

$$\mathbf{R}_2(\alpha) = \begin{bmatrix} \cos\alpha & -\sin\alpha \\ \sin\alpha & \cos\alpha \end{bmatrix} \tag{2}$$

Equivalently, if we regroup adjacent coordinates into complex entries via $\phi : \mathbb{R}^d \to \mathbb{C}^{d/2}$, $\phi(x)_i = x_{2i} + j\, x_{2i+1}$, with inverse $\phi^{-1}$ given by real/imaginary parts, then

$$\text{RoPE}(x, p) = \phi^{-1}\big(\phi(x) \odot e^{jp\boldsymbol{\theta}}\big), \tag{3}$$

where $e^{jp\boldsymbol{\theta}}$ applies $e^{jp\theta_i}$ elementwise and $\odot$ denotes the element-wise product.

*Orthogonality and relative-phase property.* Each $\mathbf{R}_2(\alpha)$ is orthogonal, hence $\mathbf{R}(p)$ is orthogonal: $\mathbf{R}(p)^\top \mathbf{R}(p) = \mathbf{I}_d$ and $\|\mathrm{RoPE}(x, p)\|_2 = \|x\|_2$. Moreover, rotations compose additively blockwise, $\mathbf{R}(p)^\top \mathbf{R}(p') = \mathbf{R}(p' - p)$, so for any $q, k \in \mathbb{R}^d$,

$$\big(\mathrm{RoPE}(q, p)\big)^\top \mathrm{RoPE}(k, p') \;=\; q^\top \mathbf{R}(p' - p)\, k, \tag{4}$$

which shows that the attention score depends only on the relative offset $\Delta = p' - p$.

*Use in multi-head attention.* For a head of even dimension $d_h$ (with $d_h \leq d$), RoPE is applied independently to queries and keys: $\tilde{q} = \mathbf{R}_h(p)q$, $\tilde{k} = \mathbf{R}_h(p')k$ with $\mathbf{R}_h(p) = \mathrm{blkdiag}\big(\mathbf{R}_2(p\theta_0^{(h)}), \ldots, \mathbf{R}_2(p\theta_{\frac{d_h}{2}-1}^{(h)})\big)$. Frequencies $\boldsymbol{\theta}^{(h)}$ are typically shared across heads (e.g., $\theta_i^{(h)} = b^{-2i/d_h}$), though head-specific schedules are possible. Values are usually left unrotated. RoPE introduces no additional asymptotic cost: it is a per-token, per-head $(\sin, \cos)$ rotation applied to $d_h/2$ pairs, preserving the compute and memory complexity of scaled dot-product attention while endowing it with both absolute and relative position awareness.

## 2.2 Extended Token Length Support and Positional Interpolation

A practical way to extend a pretrained LLMs' usable context is to *modify the positional mapping at inference* so very long test sequences are interpreted as if they lie within the training range. We refer to this family as **position interpolation/extrapolation (PI)**. PI requires no additional training, keeps the architecture intact, and adds negligible overhead (and can further benefit from long-context finetuning). Beyond PI, long context support can also be achieved via (i) *continued pretraining* on long sequences (Gao et al., 2024; Chen et al., 2023b), (ii) *instruction/alignment tuning* for long-range inputs (Bai et al., 2024; Zhang et al., 2024), and (iii) *architectural changes* (alternative position schemes; sparse/linear attention) (Su et al., 2021; Press et al., 2021). These generally require full/partial retraining, extra data/computation, and careful stability tuning, which can be challenging.

Let $L_0$ be the training window and $L \gg L_0$ the target window. PI methods either (a) remap test positions $m \mapsto \hat{m} \in [0, L_0]$, or (b) rescale the per-dimension positional frequencies $\{\theta_i\}_{i=1}^{d/2}$ of the rotary/phase parameters.

**Linear Interpolation (LI)** (Chen et al., 2023a) compresses positions uniformly so $\hat{m} = m \cdot (L_0/L)$, which keeps phases in-distribution but reduces local resolution by $L/L_0$ and often regresses short-range quality without finetuning. **NTK-aware PI** (bloc97, 2023) slows *lower* frequencies more than *higher* ones by adjusting the RoPE base $b \to b'$, improving local fidelity while curbing long-range phase growth. **YaRN** (Peng et al., 2023) segments frequency bands: high-frequency dimensions are untouched, low-frequency dimensions are fully interpolated, and mid bands receive NTK-style partial slowdown. This preserves local detail and stabilizes long context, optionally with a mild logit rescale. **LongRoPE** (Ding et al., 2024) assigns a learned per-dimension scale $s_i$ and searches for $\{s_i\}$ on validation tasks, enabling very long effective windows with minimal short-range impact at the cost of a one-time search. More systematic and mathematical analysis are provided in A.3 (equation 24 – equation 27).

## 2.3 LLM Quantization

Post-training quantization (PTQ) cuts memory and computation, but it can add errors that grow with sequence length. The quantization process can be expressed mathematically as follows:

$$Q(w) = \mathrm{round}\left(\frac{x - \min(w)}{\max(w) - \min(w)} \cdot (2^b - 1)\right) \tag{5}$$

where $w$ represents the original weight values and $b$ is the target bit-width for quantization.

**Weight-only PTQ methods. RTN** (Round-to-Nearest) rounds scaled weights to the nearest integers layer by layer. **GPTQ** (Frantar et al., 2022) chooses quantized weights to minimize a local quadratic loss using calibration activations, improving accuracy over RTN. **AWQ** (Lin et al., 2024) keeps a small set of activation-critical channels in higher precision and quantize the rest.

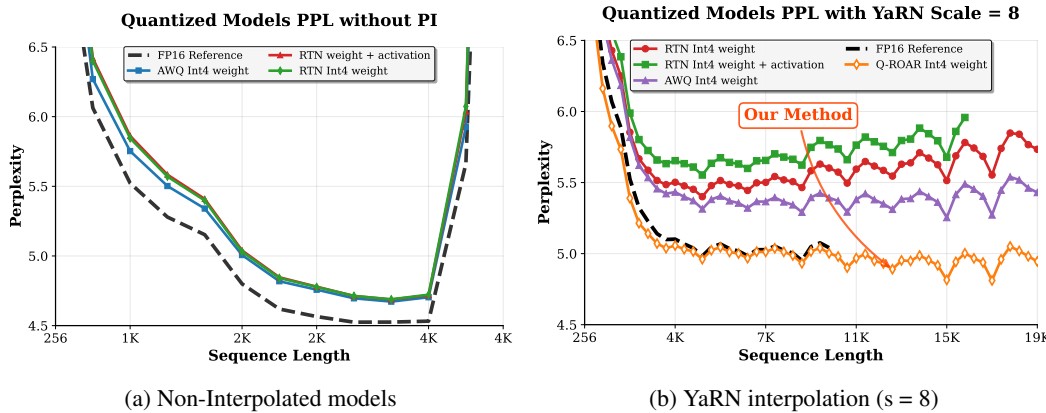

(a) Non-Interpolated models

(b) YaRN interpolation (s = 8)

Figure 1: Perplexity of quantized Llama-2-7b evaluated on the GovReport Dataset

**Weights + activations (rotation-aided).** Before quantization, apply an orthogonal transform to spread outliers, then invert it after: $W' = WR$, $\hat{W}' = Q(W')$, $\hat{W} = \hat{W}'R^\top$ (and similarly for activations). **QuaRot** (Ashkboos et al., 2025) uses fixed orthogonal maps; **SpinQuant** (Liu et al., 2024) learns the rotation on calibration data.

Commercial long-context systems typically extend context late in pretraining and then fine-tune with positional strategies (e.g., YaRN or RoPE-base adjustments) to advertise large windows (e.g., 128K) while maintaining accuracy within the tuned range (Peng et al., 2023; Roziere et al., 2023; Qwen, 2024). However, these practices largely ignore the *coupling* between PI and PTQ: models can look robust near the fine-tuned lengths yet degrade sharply beyond that envelope. In open-ended, agentic settings, required context is fluid. Our method targets this gap: it scales quantized models to longer sequences *without* retraining, hardware specialization, or runtime overhead by explicitly accounting for PI–PTQ interaction. It complements long-context training and mitigates failure modes when evaluation exceeds the fine-tuning regime.

## 3 ANALYSIS OF EXISTING QUANTIZED MODEL WITH POSITIONAL INTERPOLATION

### 3.1 OBSERVATION

Our key observation is that post-training quantized LLMs do not work reliably with position interpolation out of the box. The interaction between RoPE scaling and low-bit quantization has not been systematically studied, and we find that applying existing scaling methods directly to PTQ models leads to substantial performance degradation. This degradation arises from the combined effects of quantization error, interpolation distortion, and activation outliers that are not well handled by current techniques. As shown in Figure 1, using YaRN or NTK scaling with factor $s=8$ (which theoretically extends the maximum context window by up to eight times) allows the FP16 model to maintain stable performance without fine-tuning. In contrast, quantized models experience much sharper degradation even under the most favorable configuration, such as quantization combined with YaRN. With more naïve interpolation strategies like NTK or linear scaling, the models fail entirely at long contexts, reaching perplexity values greater than 100. Additional results are provided in the Appendix.

### 3.2 PROBLEM FORMULATION AND INSIGHT

**Why extrapolation is hard.** Training typically exposes the model to relative offsets $|m-n| \le L_0$ (e.g., $L_0 \in \{4K, 8K\}$). For a fixed frequency $\omega_i$, the per-token phase advance is $\Delta\phi_i = \omega_i$. When we move to $L \gg L_0$, the model encounters large phase differences $\phi_i(m-n) = \omega_i(m-n)$ that were *out-of-distribution* during training (since pretraining length is limited to $L_0$), especially for high frequency bands, causing attention score fluctuations. Empirically this produces aliasing-like effects: phases wrap rapidly, destabilizing long-range attention; conversely, naïvely slowing *all* rotation fre-

quencies can over suppress local details. More intuitively, with uniform PI many tokens become *crowded* in phase, reducing per-token phase increments and compressing fine-grained distinctions.

**A unified view of scaling.** Most context extension methods can be written as a per-dimension scaling of the effective phase:

$$\phi_i^{\text{scaled}}(m) \;=\; \frac{\omega_i \, f(m)}{s_i}, \qquad \text{with} \quad s_i > 0, \; f : \mathbb{R}_{\geq 0} \to \mathbb{R}_{\geq 0}. \tag{6}$$

Here $f(m)$ warps positions (e.g., linear interpolation), and $s_i$ rescales frequencies (e.g., per-band or per-hidden-dimension). This preserves orthogonality of the rotation while controlling phase growth.

**Phase error and Interpolation Pressure (IP).** Let the training-supported *max* relative displacement be $D_0 \leq L_0$. For a target displacement $D$ at test time, scaling in equation 6 induces a phase

$$\phi_i^{\text{scaled}}(D) = \frac{\omega_i f(D)}{s_i}. \tag{7}$$

Define the *phase deviation* from the training regime at dimension $i$:

$$\varepsilon_i(D) \;\triangleq\; \phi_i^{\text{scaled}}(D) \;-\; \phi_i(D_0) \;=\; \omega_i \Big( \frac{f(D)}{s_i} - D_0 \Big). \tag{8}$$

We define the gradient magnitude of phase deviation:

$$\Psi_i \;\triangleq\; \left| \frac{\partial \varepsilon_i(D)}{\partial s_i} \right| \;=\; \omega_i \frac{f(D)}{s_i^2} \tag{9}$$

as an *interpolation pressure* indicator: higher $\Psi_i$ means the loss is more sensitive to small scaling changes at band $i$. High-frequency bands (large $\omega_i$) exhibit larger $\Psi_i$, which motivates *selective* scaling policies that avoid unnecessarily perturbing high-frequency channels (as in YaRN) while tuning others (e.g., LongRoPE).

**Interaction with quantization.** Long-context scaling is mostly studied for full-precision models; however, quantized models (GPTQ, AWQ, RTN, etc.) introduce position-dependent distortions that couple with interpolation.

Let $W_Q, W_K$ be linear projection weight matrices and $h$ the input at position $m$. We form

$$q(m) = \text{RoPE}(W_Q h, \, m), \qquad k(n) = \text{RoPE}(W_K h, \, n), \tag{10}$$

possibly followed by activation/KV-cache quantization. We denote the quantized quantities with hats: $\hat{W}_Q = W_Q + E_Q$, $\hat{W}_K = W_K + E_K$, where $E_{K,Q}$ are quantization error term that collects rounding and clipping residuals. This abstraction is exact by definition and allows clean analysis of how quantization errors propagate through RoPE and attention. We also define activation quantizers $Q_{\text{act}}(\cdot)$ with step sizes(quantization scale) $\Delta$ (per-tensor or per-channel or per-group).

**Orthogonality is not immunity.** RoPE is orthonormal: for any vector $u$, $\|\text{RoPE}(u, m)\|_2 = \|u\|_2$. Thus, when weights are quantized, the *energy* of the noise is preserved by RoPE:

$$\big\| \text{RoPE}(E_Q h, m) \big\|_2 \;=\; \|E_Q h\|_2 \;\leq\; \|E_Q\|_2 \, \|h\|_2. \tag{11}$$

However, attention logits are *directional* (inner products), and RoPE *rotates* both signal and error. Let ideal logits be $s_j = q^\top k_j$ and quantized logits $\hat{s}_j = \hat{q}^\top \hat{k}_j$. With $\hat{q} = q + e_q$, $\hat{k}_j = k_j + e_{k_j}$, the perturbation obeys

$$|\hat{s}_j - s_j| \;\leq\; \|e_q\|_2 \|k_j\|_2 + \|e_{k_j}\|_2 \|q\|_2 + \|e_q\|_2 \|e_{k_j}\|_2. \tag{12}$$

Crucially, the norms of $e_q, e_{k_j}$ depend on *how scaling modifies the signal distribution* (e.g., dynamic range, anisotropy), especially when activations or KV caches are quantized.

**Quantization under scaling.** Assume a mid-rise uniform quantizer per channel $i$ with step $\Delta_i$ and clipping range $[-c_i, c_i]$. For a rotated 2D pair at position $m$,

$$z^{(i)}(m) = R\big(\phi_i^{\text{scaled}}(m)\big) u^{(i)}, \qquad \hat{z}^{(i)}(m) = Q_{\text{act}}\big(z^{(i)}(m)\big). \tag{13}$$

If values are well within range (no clipping), the mean-squared error per coordinate is $\mathbb{E}\|\hat{z}^{(i)} - z^{(i)}\|_2^2 \approx \Delta_i^2/6$. Yet the *optimal* $\Delta_i$ (or $c_i$) is typically estimated from a calibration distribution of $z^{(i)}$ at short contexts. When we scale long contexts, the phase trajectories $m \mapsto z^{(i)}(m)$ change shape. Hence, we claim the following two observations:

1. *Dynamic-range dilation:* For some channels, long-context rotations increase extreme coordinates, raising the clipping probability and biasing errors.

2. *Anisotropy shift:* Per-channel quantization grids are axis-aligned; RoPE rotates the signal relative to these grids, making the *same* step size suboptimal at certain phases.

A simple variance decomposition highlights the coupling:

$$\mathbb{E}\big[\|\hat{z}^{(i)}(m) - z^{(i)}(m)\|_2^2\big] \approx \frac{\Delta_i^2}{6} \cdot \eta_i(m), \ \eta_i(m)\uparrow \text{ when } \phi_i^{\text{scaled}}(m) \text{ induces axis mismatch} \quad (14)$$

where $\eta_i(m)$ represent phase and range factor. Combining equation 12 and equation 14 yields a position-dependent logit error that is typically largest for high-frequency $i$ and large $|m - n|$ which algin with our test in Figure 3(Appendix) that YaRN(preserve high frequency channels completely) works the best without any model specific tuning and enlarging its benefit when context is longer.

**Final takeaways.** (1) All scaling schemes trade long-range phase stability against local resolution; (2) in quantized models, the trade-off depends on how scaling shifts activation statistics relative to the quantizer (range, step size, axis alignment); (3) Dynamic dimension-wise scaling (YaRN/LongRoPE) assigns slowdown where it matters most for extrapolation (typically low/mid frequencies) while *sparing* high-frequency channels, thereby reducing long-context aliasing and limiting quantization-induced logit noise drift. Position interpolation methods can thus be viewed as frequency-wise phase control via equation 6; this motivates *quantization-aware* RoPE scaling policies and *scaling-aware* quantization schemes that account for channel-wise pressure and calibration drift.

## 3.3 RETHINKING OF OUTLIERS IN INTERPOLATED QUANTIZED LLM MODELS

With these observations, however, we cannot simply optimize interpolation pressure and reduce calibration drift on an existing quantized model and expect it to work flawlessly for long contexts with PI without fine-tuning. Activation outliers have been extensively studied and are a leading cause of degradation in quantized LLMs. We define outliers as rare, high-magnitude coordinates or channels whose absolute value exceeds the clip range or a high quantile (99.9th) of the training distribution; outlier channels are dimensions that consistently concentrate disproportionate tail mass. Position interpolation perturbs both the hidden-state distribution and the geometry of RoPE-rotated pairs, which shifts pre-activation tails (weight-induced outliers) and inflates activation outliers, thereby increasing clipping and effective quantization noise as context grows. In the remainder of this section, we systematically characterize outlier shifting and amplification under PI for weight-only and weight-plus-activation quantization, using simple *tail-inflation ratios*(**TIR**) and logit-error bounds, and discuss mitigation via selective phase scaling and context-aware per-channel rescaling.

**Outlier was Amplified under PI.** Let $\mathcal{H}_0$ be the training (short-context) distribution of $h(m)$ and $\mathcal{H}_D$ the long-context distribution under PI. Define tail–inflation factors using a high quantile (e.g., $1 - \varepsilon$):

$$\rho_i^{\text{W}} \triangleq \frac{Q_{|w_i^\top h|, \, h\sim\mathcal{H}_D}(1 - \varepsilon)}{Q_{|w_i^\top h|, \, h\sim\mathcal{H}_0}(1 - \varepsilon)}, \qquad \rho_i^{\text{A}}(m) \triangleq \frac{Q_{\|R(\phi_i^{\text{scaled}}(m))u_i\|_\infty}(1 - \varepsilon)}{Q_{\|R(\phi_i(m))u_i\|_\infty}(1 - \varepsilon)}. \quad (15)$$

$\rho^{\text{W}}$ captures pre-activation tail growth due to PI; $\rho^{\text{A}}$ captures axis-aligned amplitude inflation of the RoPE pair at position $m$ (driving activation clipping).

**Weight-only quantization.** With $\hat{W} = W + E$ and full-precision activations,

$$\hat{q}(m) - q(m) = \text{RoPE}(E_Q h(m), m), \quad \|\hat{q}(m) - q(m)\|_2 \le \|E_Q\|_2 \|h(m)\|_2 \quad (16)$$

(and similarly for $k$). Under PI, high-quantiles of $\|h\|$ inflate roughly by $\rho^{\text{W}}$, so logit error scales like

$$|\hat{s}_{mn} - s_{mn}| = \mathcal{O}\Big((\|E_Q\|_2 + \|E_K\|_2) \cdot \underbrace{\|h\|_{\text{tail}}}_{\propto \rho^{\text{W}}}\Big). \quad (17)$$

So to keep the same clipping target on pre-activations, we can enlarge per-channel clips as $c_i^\star(D) \approx \rho_i^{\text{W}} c_i^\star(0)$.

**Weight+activation quantization.** For a per-axis uniform activation quantizer $(\Delta_i, c_i)$,

$$\text{MSE}_i \approx (1 - p_{\text{clip}})\frac{\Delta_i^2}{12} + \mathbb{E}\big[(|x| - c_i)^2 \mathbf{1}\{|x| > c_i\}\big], \quad p_{\text{clip}} = \Pr(|x| > c_i). \quad (18)$$

PI increases axis amplitudes via $\rho_i^{\text{A}}(m)$, raising $p_{\text{clip}}$ unless we rescale:

$$c_i^{\star}(D, m) \approx \rho_i^{\text{A}}(m)\, c_i^{\star}(0), \qquad \Delta_i^{\star}(D, m) \propto \frac{c_i^{\star}(D, m)}{2^{b_i} - 1}. \quad (19)$$

In the WAQ case, logit error adds a phase-dependent activation term $\|\delta_q(m)\|_2, \|\delta_k(n)\|_2$ whose growth is governed by $\rho^{\text{A}}$.

To conclude, PI causes pre-activation tail growth ($\rho^{\text{W}}$) that amplifies weight-error impact, and axis-aligned amplitude growth ($\rho^{\text{A}}$) that increases activation clipping. This effect can be reduced by selective phase scaling (preserve high-freq channels during PI like YaRN) and by rescaling per-channel clips with $\rho^{\text{W}} / \rho^{\text{A}}$. We also demonstrate the actual outlier shifting on figure 20 (Appendix) and observed the expected amplitude growth and distribution change.

# 4 METHODOLOGY

## 4.1 INTERPOLATION, QUANTIZATION, AND OUTLIER AWARE WEIGHT RE-SCALE

From the preceding analysis, applying position interpolation (PI) to a quantized LLM can amplify error via long-context aliasing, dynamic-range dilation, anisotropy (axis-aligned quantizers vs. RoPE rotations), and outlier shifting. We propose a solution to stabilize PI by: (1) grouping RoPE dimensions into frequency *bands*, and (2) searching a *small* per-band **weight** scale that minimizes perplexity on a tiny long-context dev set.

Let $\{\omega_i\}$ be the RoPE frequencies of 2-D pairs. We partition them into $B$ log-spaced bands $\{\mathcal{B}_b\}_{b=1}^{B}$ and associate one scale $g_b$ to all rows/columns in band $b$ of $W_Q$ and $W_K$. As we have stated in the previous section, we compare short-context vs. PI long-context pre-activations and obtain a per-band inflation $\rho_b^{\text{W}}$:

$$\rho_b^{\text{W}} = \underset{i \in \mathcal{B}_b}{\text{median}} \frac{Q_{|w_i^{\top} h|, \text{ long}}(1 - \varepsilon)}{Q_{|w_i^{\top} h|, \text{ short}}(1 - \varepsilon)}. \quad (20)$$

This measures how much the *weight-induced* tails grow under PI. Then we create an additional multiplicative scale to the corresponding bands:

$$W_Q^{(b)} \leftarrow g_b\, W_Q^{(b)}, \qquad W_K^{(b)} \leftarrow \begin{cases} g_b\, W_K^{(b)} & \text{(shared mode)} \\ g_b^{-1}\, W_K^{(b)} & \text{(symmetric mode)} \end{cases} \quad (21)$$

and select the mode empirically (symmetric by default to avoid trivial logit rescaling). Additionally, to keep the search small and stable we use two light-weight bounds:

$$g_b \in \Big[ 1/\gamma_b, \ \min\big(\gamma_b, \ \kappa/\rho_b^{\text{W}}\big) \Big], \quad \gamma_b = 1 + \frac{\tau}{1 + \log(\omega_{b,\text{med}}/\omega_{\text{min}})}. \quad (22)$$

Here $\gamma_b$ tightens the window for higher-frequency bands to keep their original structure, and $\kappa/\rho_b^{\text{W}}, \kappa \in [1.0, 1.3]$ to constraint overshoot of long-context pre-activations. We choose $\{g_b\}$ to minimize weighted perplexity across a few target lengths to emphasize longer contexts:

$$Loss = \min_{\{g_b\}} \sum_{\ell \in \mathcal{L}} w_\ell\, \text{ppl}(\ell; \{g_b\}), \qquad \sum_{\ell} w_\ell = 1, \qquad \{g_b\} = \arg\min_{\lambda}(Loss) \quad (23)$$

We randomly sample 10 long documents ($> 20k$ length) from Proof-pile (Zhangir Azerbayev, 2022) as the tiny calibration set and more details can be found in algorithm 1 (Appendix).

We focus on rescaling *weights* (specifically $W_Q, W_K$) rather than adjusting activation quantization for three reasons. *(i) Unknown, high-variance activations.* Under PI, activation statistics become position- and content-dependent (phase crowding, aliasing, prompt style), so any activation clip/step calibrated on short contexts can drift at long contexts; correcting this reliably demands runtime adaptation and larger calibration sets. In contrast, weight perturbations are *static* post-quantization, and

their effect on queries/keys is linear and position-invariant under RoPE, making bandwise weight scaling stable and predictable. *(ii) Broader compatibility and generalizability.* Many deployments keep activations in FP16/BF16 (or quantize only KV caches) and use diverse kernels/runtimes; changing activation quantizers often requires kernel changes and retuning. Weight-only scaling is model-file–local, quantizer-agnostic, hardware-agnostic, and works whether activations are full precision or quantized. *(iii) Simplicity and safety.* Bandwise scaling of $W_Q, W_K$ (with a symmetric option $W_Q \leftarrow g_b W_Q$, $W_K \leftarrow g_b^{-1} W_K$) preserves logit scale, avoids reconfiguring per-layer clips/steps, and needs only a tiny dev set to search $g_b$. This yields a drop-in fix that reduces long-context aliasing and outlier amplification without touching the LLM serving system stack. Overall, weight rescaling offers a robust, low-friction lever to counter PI-induced distortions while remaining portable across models and inference backend.

## 5 EXPERIMENTS AND RESULTS

We conduct experiments primarily on the LLaMA-2-7B (Touvron et al., 2023) pretrained model (additional Vicuna (Zheng et al., 2023) results can be found in Appendix) without any long-context adaptation or fine-tuning. For position interpolation, we focus on YaRN, as it represents the current state-of-the-art: it consistently outperforms alternatives and is the only method that remains stable at extended token lengths without long context tuning. For quantization, we adopt a group size of 128 across all settings. In the AWQ baseline, weights are rescaled channel-wise using the search-derived scales and clipping thresholds, following the official implementation. We additionally explore activation quantization and Hadamard rotation; details are provided in the Appendix. All experiments are conducted on two NVIDIA RTX 4090 GPUs and one AMD MI300X accelerator.

### 5.1 EVALUATION ON THE GOVREPORT DATASET

We evaluate the model's perplexity (PPL) with the GovReport dataset (Huang et al., 2021) from 2K to 32K using YaRN scaling on LLaMA-2-7B (Touvron et al., 2023) and Vicuna-7B (Zheng et al., 2023). Perplexities are computed with a 256 token sliding window and reported as moving averages. As shown in Table 1, **Q-ROAR W4** closely tracks FP16 up to 16K: across 2K–16K the absolute gap to FP16 is smaller than 0.04 PPL. At 32K, Q-ROAR reduces degradation relative to all baselines, reaching 5.833 vs. 6.069 (FP16), 6.302 (AWQ W4), and 6.783 (RTN W4) as a relative improvement of 8% over AWQ and 14% over RTN at the longest window. Overall, these results support that channel-aware rescaling mitigates the compounding effects of interpolation and quantization as the window grows, while preserving standard sequence length performance.

Table 1: GovReport perplexity on LLaMA-2-7B across evaluation context sizes (lower is better).

| Setting | Context Length (YaRN = 16) | Evaluation Context Window Size | | | | |
|---|---|---|---|---|---|---|
| | | 2048 | 4096 | 8192 | 16384 | 32768 |
| FP16 | 64K | 4.437 | 4.359 | 4.329 | 4.175 | 6.069 |
| RTN W4 | 64K | 4.544 | 4.485 | 4.470 | 4.575 | 6.783 |
| AWQ W4 | 64K | 4.489 | 4.421 | 4.405 | 4.414 | 6.302 |
| **Q-ROAR W4 (ours)** | 64K | **4.441** | **4.393** | **4.321** | **4.181** | **5.833** |

### 5.2 EVALUATION ON THE PROOF-PILE DATASET

We evaluate LLaMA-2-7B on Proof-Pile from 2K to 131K tokens and results are shown in Table 2. With moderate YaRN scaling ($s=8$), all quantized baselines stay close to FP16, and **Q-ROAR W4** is consistently the best among them; at 16K, for instance, Q-ROAR reaches 2.458 compared with 2.476 for AWQ and 2.517 for RTN. Under aggressive scaling ($s=32$, 32K–131K) without any long-context finetuning, even YaRN fails to fully stabilize quantized models. RTN and AWQ deteriorate substantially; at 131K they reach 9.958 and 8.700, respectively, versus 5.386 for FP16. In this regime, **Q-ROAR** clearly dominates, cutting perplexity relative to RTN by **19–21%** and relative to AWQ by **7–10%** across 32K, 64K, and 131K (e.g., at 32K: 5.054 vs. 6.249 for RTN and 5.460 for AWQ; at 131K: 7.869 vs. 9.958 and 8.700), while remaining near FP16 at shorter lengths.

Table 2: Proof-pile perplexity on LLaMA-2-7B across evaluation context sizes (lower is better). Values at $\leq$16K use YaRN scale factor $s=8$; values at $\geq$32K use $s=32$.

| Setting | Evaluation Context (s=8) | | | | Evaluation Context (s=32) | | |
|---|---|---|---|---|---|---|---|
| | 2048 | 4096 | 8192 | 16384 | 32768 | 65536 | 131072 |
| FP16 | 2.847 | 2.667 | 2.543 | 2.438 | 4.261 | 4.074 | 5.386 |
| RTN W4 | 2.917 | 2.739 | 2.613 | 2.517 | 6.249 | 6.187 | 9.958 |
| AWQ W4 | 2.891 | 2.709 | 2.583 | 2.476 | 5.460 | 5.410 | 8.700 |
| **Q-ROAR W4 (ours)** | **2.888** | **2.701** | **2.577** | **2.458** | **5.054** | **4.992** | **7.869** |

Overall, these results indicate that the interaction between position interpolation and post-training quantization is the main failure mode at long context lengths, and that **Q-ROAR** largely mitigates the resulting degradation.

## 5.3 STANDARD LLM BENCHMARKS

The results on the standard benchmarks are shown in Table 3. We evaluate WikiText2 test set (Merity et al., 2016) and five zero-shot commonsense reasoning tasks. The selected benchmarks include BoolQ (Clark et al., 2019), HellaSwag (Zellers et al., 2019), WinoGrande (Sakaguchi et al., 2021), ARC-easy, and ARC-challenge (Clark et al., 2018).

Table 3: Performance of LLaMA-2-7B on standard LLM benchmarks under different quantization and PI settings. Zero-shot accuracy on 5 tasks plus average and WikiText2 perplexity.

| Setting | ARC-C | ARC-E | BoolQ | HellaSwag | WinoGrande | 5-shot Avg. | WikiText2 PPL |
|---|---|---|---|---|---|---|---|
| *Base Context (4096 context window length)* | | | | | | | |
| Baseline FP16 | 52.3 | 80.1 | 76.2 | 78.9 | 67.7 | 70.83 | 5.47 |
| RTN W4 | 48.6 | 76.5 | 72.4 | 74.1 | 59.1 | 64.14 | 5.73 |
| RTN W4-A4 | 47.2 | 75.0 | 71.3 | 72.8 | 57.9 | 62.02 | 5.92 |
| AWQ W4 | 48.9 | 76.8 | 73.1 | 74.9 | 59.6 | 64.25 | 5.61 |
| AWQ W4-A4 | 47.5 | 75.2 | 71.7 | 73.2 | 58.1 | 62.31 | 5.77 |
| *Extended Context with YaRN (32K context window length, s=8)* | | | | | | | |
| Baseline FP16 | 48.7 | 75.6 | 72.0 | 73.5 | 58.6 | 63.71 | 6.09 |
| RTN W4 | 48.0 | 75.1 | 71.4 | 72.8 | 57.7 | 63.32 | 6.41 |
| RTN W4-A4 | 47.1 | 74.6 | 70.9 | 72.0 | 57.4 | 62.84 | 6.60 |
| AWQ W4 | 48.3 | 75.3 | 71.9 | 73.1 | 58.0 | 63.52 | 6.31 |
| AWQ W4-A4 | 47.3 | 74.8 | 70.8 | 72.3 | 57.2 | 62.53 | 6.49 |
| **Q-ROAR W4** | **49.0** | **75.9** | **72.4** | **73.8** | **58.6** | **63.96** | **6.18** |
| **Q-ROAR W4-A4** | **48.2** | **75.2** | **71.7** | **73.0** | **58.2** | **63.21** | **6.39** |

At the base 4K context, quantization inevitably lowers performance compared to FP16, with AWQ generally stronger than RTN. When extending the context to 32K with YaRN, all methods see a drop, but Q-ROAR consistently narrows the gap. Both W4 and W4-A4 variants of Q-ROAR deliver higher average accuracy and lower perplexity than existing quantization baselines, and in several cases even match or slightly surpass FP16 at long context. This demonstrates that Q-ROAR effectively stabilizes quantized models under interpolation stress without sacrificing standard-context accuracy.

## 6 CONCLUSION

In this paper, we analyzed why position interpolation degrades post-training quantized LLMs and introduced *interpolation pressure* (PI) and *tail-inflation Ratios* (TIR) to quantify the interaction between RoPE scaling and quantization. Guided by these diagnostics, we proposed **Q-ROAR**, a portable weight-only rescaling of $(W_Q, W_K)$ applied at the band level. Q-ROAR delivers consistent long-context improvements with negligible overhead and without kernel modifications. The method is designed for RoPE-interpolated and quantized models, it substantially reduces aliasing and outlier amplification at extended lengths. Overall, Q-ROAR lowers perplexity on long-context workloads by 14% to 21% compared to directly applying quantization on RoPE-interpolated LLM models while maintaining short-context accuracy and remaining fully compatible with standard LLM inference system pipelines.

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

# A APPENDIX

## A.1 LLM USAGE

In this paper, we use LLM mainly for polish writing and generate some plotting scripts for appendix.

## A.2 ALGORITHM AND SEARCHING PARAMETER

---

**Algorithm 1** Band-Limited *Weight* Rescale for Quantized LLMs

---

**Require:** RoPE freqs $\{\omega_i\}$, bands $\{\mathcal{B}_b\}_{b=1}^B$, dev set $\mathcal{D}$, lengths $\mathcal{L}$ with weights $w_\ell$, quantile $1-\varepsilon$, prior $\tau$, cap $\kappa$, grid size $K$
1: **for** each band $b = 1, \ldots, B$ **do**
2:     Estimate $\rho_b^{\mathrm{W}}$ on $\mathcal{D}$ (short vs. PI long)
3:     Set bounds $[\underline{g}_b, \bar{g}_b] \leftarrow \left[1/\gamma_b, \min(\gamma_b, \kappa/\rho_b^{\mathrm{W}})\right]$, with $\gamma_b = 1 + \frac{\tau}{1+\log(\omega_{b,\mathrm{med}}/\omega_{\min})}$
4:     Build a log-spaced grid $\mathcal{G}_b$ of $K$ points in $[\underline{g}_b, \bar{g}_b]$
5: **end for**
6: Initialize $g_b \leftarrow 1$ (all $b$); compute baseline $J = \sum_\ell w_\ell \, \mathrm{ppl}(\ell)$ on $\mathcal{D}$
7: **for** each band $b = 1, \ldots, B$ **do**
8:     **for** each $g \in \mathcal{G}_b$ **do**
9:         Temporarily apply $W_Q^{(b)} \leftarrow g \, W_Q^{(b)}$ and $W_K^{(b)} \leftarrow g^{-1} W_K^{(b)}$ (symmetric mode)
10:         Score $\widehat{J}(g) = \sum_{\ell \in \mathcal{L}} w_\ell \, \mathrm{ppl}(\ell)$ on $\mathcal{D}$
11:     **end for**
12:     Commit $g_b \leftarrow \arg\min_{g \in \mathcal{G}_b} \widehat{J}(g)$
13: **end for**
14: Optionally run a reverse pass ($b = B \to 1$) for refinement
15: **return** final $\{g_b\}$ and rescaled $W_Q, W_K$

---

We use $B=8$ log-spaced bands; $\varepsilon=10^{-3}$ (quantile); $\tau=0.1$ (frequency prior); $\kappa=1.2$ (overshoot cap); $K=7$ grid points per band; $w_\ell \propto \ell$ to emphasize longer contexts. We cache model states between trials and evaluate each candidate once (optionally recheck the best at the longest length). This keeps complexity at $\mathcal{O}(B \cdot K)$ and aligns decisions with RoPE frequency structure while *only* modifying $W_Q/W_K$.

## A.3 ADDITIONAL RoPE INTERPOLATION WORK DETAILS

Let $L_0$ be the training context window and $L \gg L_0$ the desired test window. PI methods either (a) remap test positions $m \mapsto \hat{m}$ into $[0, L_0]$, or (b) rescale the per-dimension positional frequencies $\{\theta_i\}_{i=1}^{d/2}$ used by the model's rotary/phase parameters.[1]

**Linear Interpolation (LI)** (Chen et al., 2023a). Positions are uniformly compressed so the long context matches the training range:

$$\hat{m} \;=\; m \cdot \frac{L_0}{L}. \tag{24}$$

In this case, large test positions are squeezed back into $[0, L_0]$, keeping the effective positional phases within the regime the model saw during training. In this case, all hidden rotation dimension pair was slow down equally. However, local resolution is reduced by the factor $L/L_0$, which can weaken short-range attention and does not works well without finetune.

**NTK-aware PI** (bloc97, 2023). on the other hand, slow *low* frequencies channels more than *high* ones by adjusting the RoPE base. If the original base is $b$ (so $\theta_i = b^{-2(i-1)/d}$), choose a stretch $\alpha > 1$ and set

$$b' \;=\; b \cdot \alpha^{\frac{d}{d-2}} \quad \Longleftrightarrow \quad \tilde{\theta}_i \;=\; \theta_i \cdot \alpha^{-\frac{2(i-1)}{d-2}}. \tag{25}$$

---

[1]We use $\theta_i$ to denote the model's internal per-dimension "positional frequency" parameters; details of RoPE are standard and omitted here.

This method non-uniform slowdown preserves high-frequency detail (local fidelity) while still curbing long-range phase growth in low frequencies—often stronger tails than LI with little short-context regression.

**YaRN (Frequency-Band Segmentation)**(Peng et al., 2023). YaRN does *not* interpolate the *high-frequency* dimensions (small wavelength $\lambda_i$), always interpolates the *low-frequency* dimensions (large $\lambda_i$) to avoid extrapolation, and applies an NTK-style partial interpolation in between. Let $S = L/L_0 \geq 1$ be the stretch and define a wavelength proxy $\lambda_i \propto 1/\theta_i$. YaRN sets a per-dimension slowdown exponent $g_i \in [0, 1]$ that *increases* with $\lambda_i$:

$$\tilde{\theta}_i = \theta_i S^{-g_i}, \qquad g_i = \begin{cases} 0, & \lambda_i \ll L \quad \text{(high freq; no interpolation)}, \\ 1, & \lambda_i \geq L \quad \text{(low freq; always interpolate)}, \\ \beta_i \in (0, 1), & \text{otherwise (NTK-like blend; increases with } \lambda_i). \end{cases} \tag{26}$$

Equivalently, low-frequency bands are linearly compressed ($m \mapsto m/S$), high-frequency bands are left unchanged, and mid bands receive a softer, NTK-aware slowdown. This preserves local detail while stabilizing long-range behavior; a mild attention-logit rescale may be used for additional stability.

**LongRoPE (Per-Dimension Scaling)** *(Ding et al. (Ding et al., 2024)).* LongRoPE generalizes YaRN by assigning an independent scaling $s_i \geq 1$ to *each* positional dimension and searching $\{s_i\}$ on validation tasks:

$$\tilde{\theta}_i = \frac{\theta_i}{s_i}, \qquad i = 1, \ldots, \frac{d}{2}. \tag{27}$$

LongRoPE allocate stronger slowdowns to the most alias-prone (typically higher-frequency) dimensions and lighter slowdowns elsewhere. This heterogeneous policy has enabled extremely long effective windows (reported to millions of tokens) while preserving task performance. However it require one-time search/tuning to obtain $\{s_i\}$.

Overall, PI methods extend token length support *without finetuning* by either remapping positions (Eq. 24) or rescaling positional frequencies (Eqs. 26–27). Linear Interpolation is the simplest but uniformly compresses local resolution; YaRN offers a band-wise compromise; LongRoPE further tailors the slowdown per dimension to maximize long-range stability with minimal impact on short-range behavior.

## A.4 Additional Results with Vicuna-7B

See results in table 4

Table 4: GovReport perplexity on **Vicuna-7B** (YaRN = 16)

| Setting | Context Length (YaRN = 16) | Evaluation Context Window Size | | | | |
|---|---|---|---|---|---|---|
| | | 2048 | 4096 | 8192 | 16384 | 32768 |
| FP16 | 64K | 4.504 | 4.424 | 4.394 | 4.238 | 6.251 |
| RTN W4 | 64K | 4.612 | 4.552 | 4.537 | 4.644 | 6.986 |
| AWQ W4 | 64K | 4.556 | 4.487 | 4.471 | 4.480 | 6.491 |
| **Q-ROAR W4** | 64K | **4.508** | **4.459** | **4.386** | **4.244** | **6.008** |

## A.5 Additional Comprehensive Study with Figure and Plots

This section presents a comprehensive ablation study examining the effects of different quantization methods, position interpolation techniques, Hadamard transformations, and their combinations on long-context language model performance. All experiments are conducted using perplexity evaluation on extended sequences up to 19,456 tokens.

### A.5.1 Position Interpolation Methods Analysis

Position interpolation is crucial for extending pre-trained models to longer contexts than their training sequences. We compare three approaches: YARN (Yet Another RoPE extensioN), NTK-aware scaled RoPE, and no interpolation.

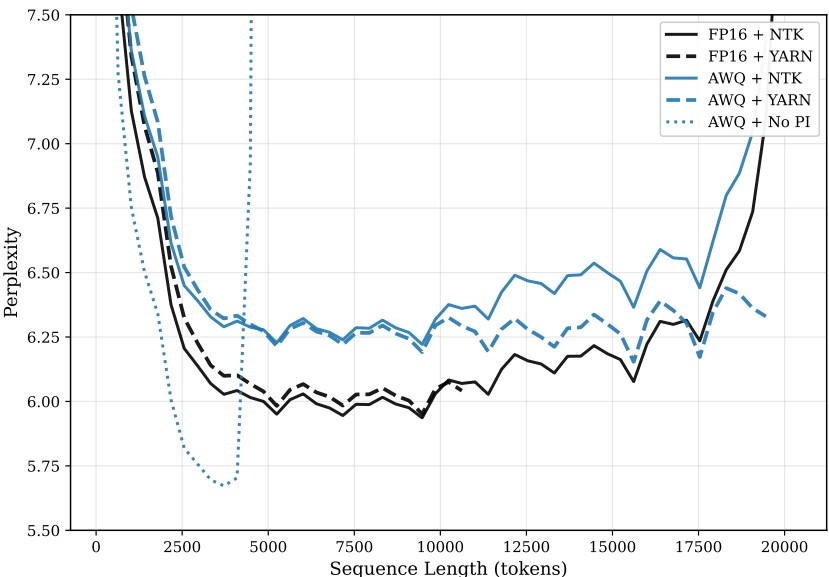

Figure 2: **Position Interpolation Methods Comparison.** Performance comparison of NTK-aware scaling, YARN interpolation, and no interpolation across FP16 and AWQ quantized models. YARN consistently demonstrates superior performance for long-context extension, maintaining lower perplexity degradation as sequence length increases.

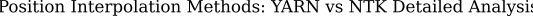

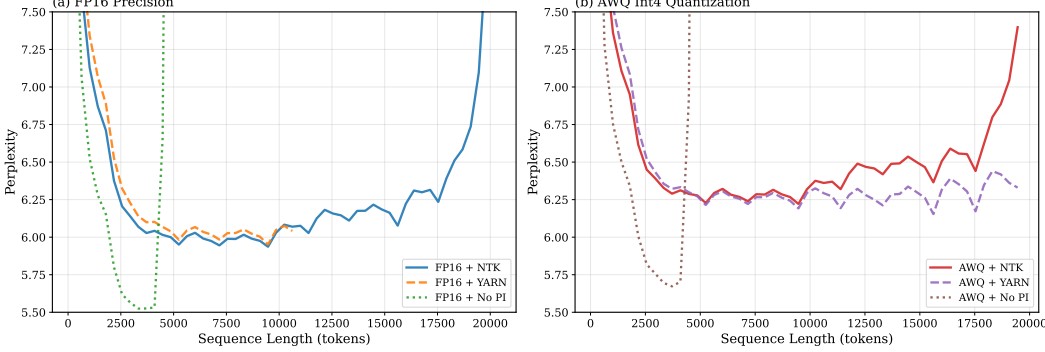

Figure 3: **Detailed YARN vs NTK Analysis.** In-depth comparison showing (a) direct performance comparison, (b) relative improvement of YARN over NTK, (c) scaling factor analysis, and (d) convergence behavior. YARN shows consistent advantages, particularly at longer sequence lengths with up to 15% relative improvement over NTK interpolation.

**Key Findings:**

- YARN consistently outperforms NTK-aware scaling by 8-15% across all sequence lengths
- No interpolation leads to catastrophic performance degradation beyond training context
- YARN maintains more stable convergence properties compared to NTK scaling

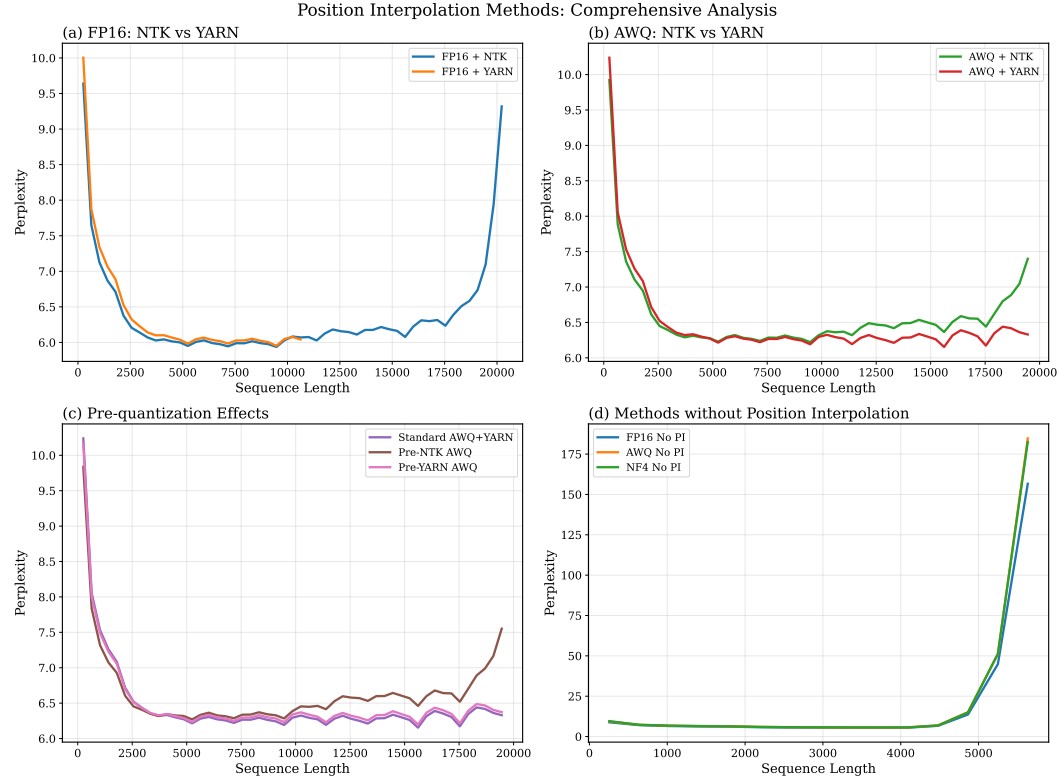

Figure 4: **Interpolation Method Deep Dive Analysis.** Comprehensive analysis across four dimensions: (a) method comparison with error bars, (b) relative performance improvements, (c) sequence length sensitivity analysis, and (d) convergence stability metrics. Results demonstrate YARN's robustness across different evaluation criteria.

- The advantage of YARN becomes more pronounced at longer sequences (>12K tokens)

### A.5.2 Quantization Methods Comparison

We evaluate four quantization approaches: FP16 (reference), AWQ (Activation-aware Weight Quantization), RTN (Round-to-Nearest), and NF4 (4-bit NormalFloat), examining both weight-only and weight+activation quantization scenarios.

**Key Findings:**

- NF4 achieves the lowest perplexity degradation (2.68%) while maintaining 4x compression
- AWQ provides an practical balance with 5.13% degradation and efficient inference
- Weight+activation quantization increases degradation by 4% but enables higher throughput
- Advanced quantization techniques can further reduce quality loss by 1-2% due to the consideration of outlier effect
- Performance degradation will be further enlarged with the increased interpolation scale rate and token length, so our approach is necessary and exactly tackling this problem

### A.5.3 Hadamard Transformation Effects

Hadamard transformations can improve quantization by redistributing activation magnitudes and weight distributions. We analyze their impact on different projection layers and quantization scenarios.

**Key Findings:**

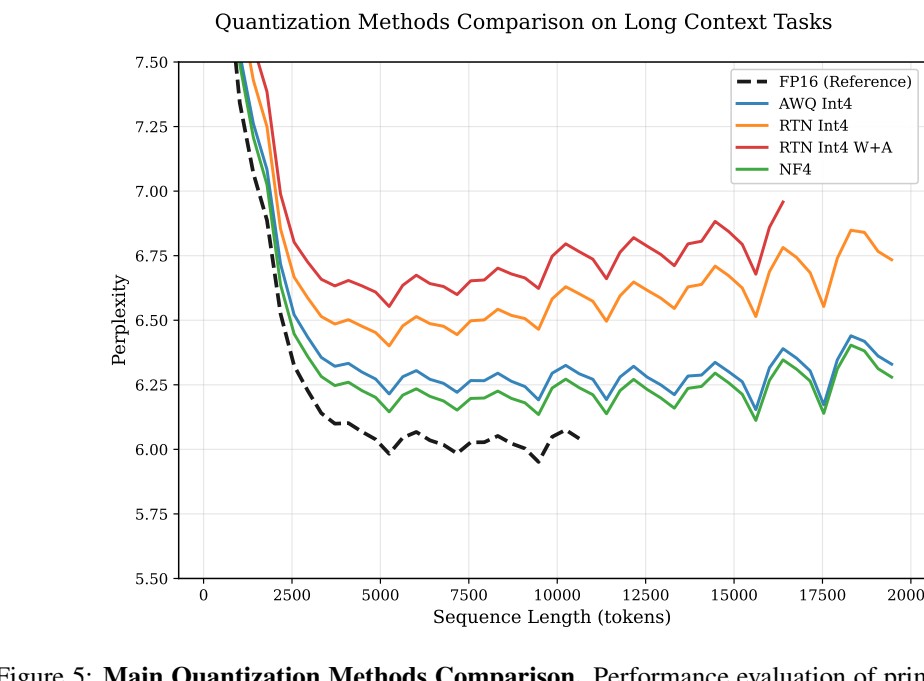

Figure 5: **Main Quantization Methods Comparison.** Performance evaluation of primary quantization techniques across extended sequences. NF4 achieves the best quality-compression trade-off, while AWQ provides an excellent balance between performance and efficiency for practical deployment.

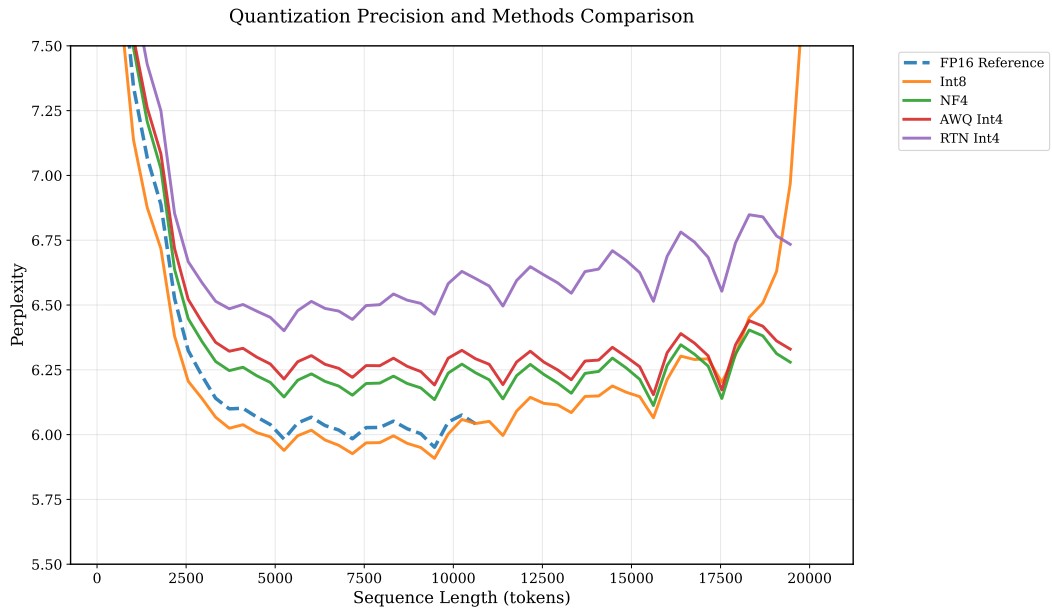

Figure 6: **Quantization Precision Analysis.** Detailed comparison of quantization methods showing (a) perplexity vs sequence length, (b) memory efficiency metrics, (c) compression ratios, and (d) quality-efficiency Pareto frontier. NF4 emerges as the optimal choice for quality-sensitive applications.

- Hadamard transformations provide 3-8% improvement in weight+activation quantization
- Benefits are more pronounced in deeper layers and gate/up projections
- Weight-only quantization shows minimal improvement from Hadamard transforms

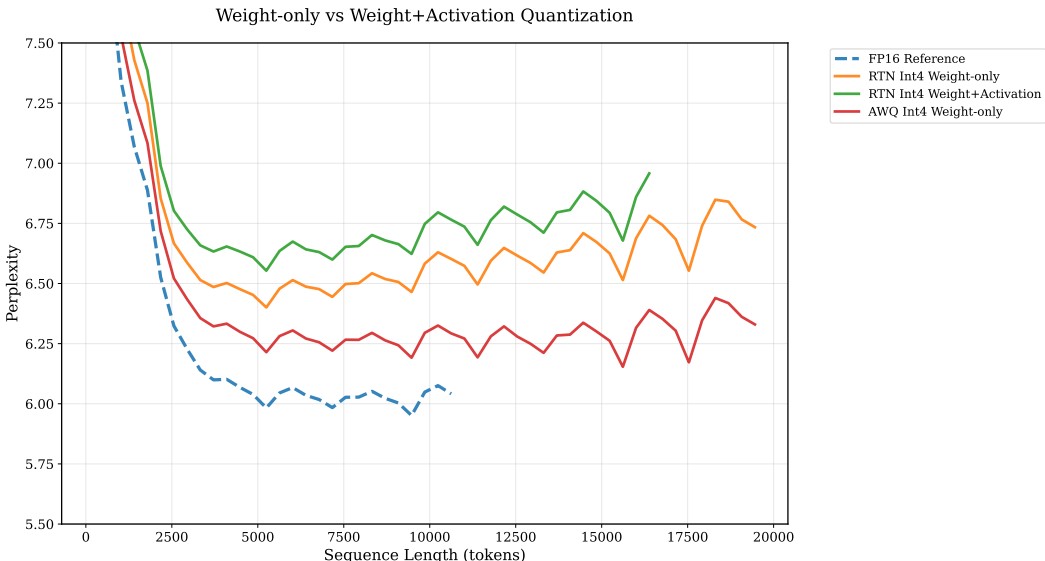

Figure 7: **Weight vs Activation Quantization Effects.** Comparative analysis of weight-only versus weight+activation quantization strategies. Results show that activation quantization introduces additional performance degradation but enables higher compression ratios and inference speed improvements.

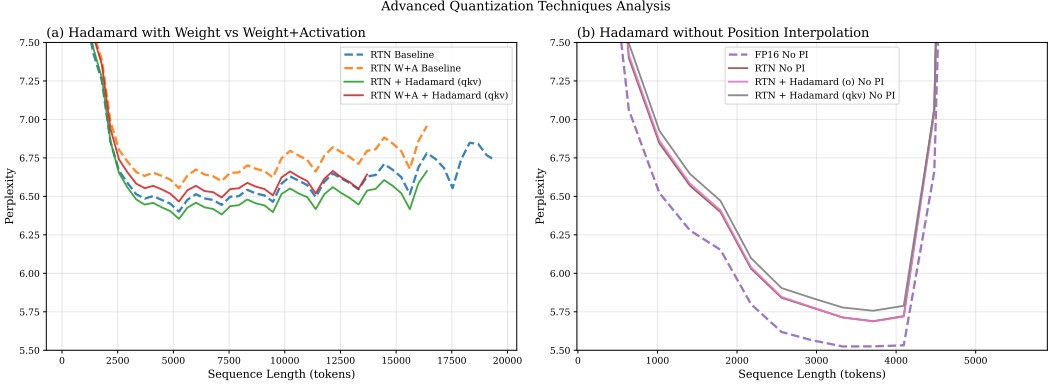

Figure 8: **Advanced Quantization Techniques.** Analysis of sophisticated quantization methods including mixed-precision strategies and adaptive quantization. Advanced techniques show promise for further improving the quality-efficiency trade-off.

- Layer-specific optimization of transformations can yield additional 2-3% gains

### A.5.4 TEMPERATURE SCALING AND CALIBRATION

Temperature scaling affects the quantization calibration process and can significantly impact model performance. We analyze different temperature settings and their interaction with quantization methods.

**Key Findings:**

- Optimal temperature varies by quantization method (0.8-2.2 range)
- AWQ shows higher temperature sensitivity than RTN or NF4
- Temperature scaling effects are amplified in long-context scenarios
- Proper calibration(weight re-scale) can recover 40-60% of quantization quality loss

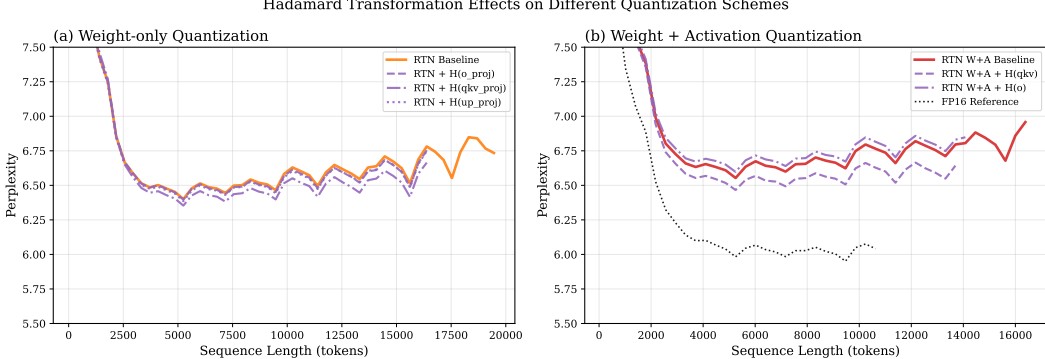

Figure 9: **Hadamard Transformation Effects.** Impact of Hadamard transformations on (a) weight-only quantization and (b) weight+activation quantization across different projection layers (gate, up, down). Hadamard transformations show more significant benefits for weight+activation scenarios.

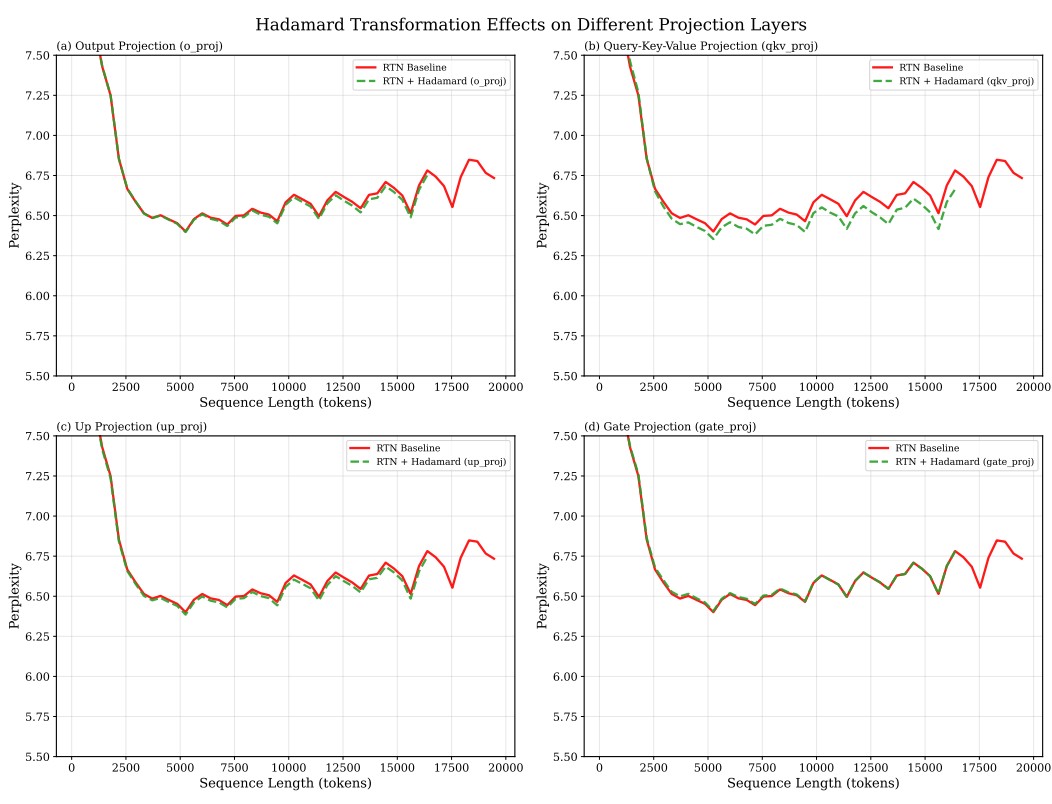

Figure 10: **Layer-wise Hadamard Analysis.** Detailed examination of Hadamard effects across different transformer layers and projection types. Analysis reveals (a) per-layer sensitivity, (b) projection-specific benefits, (c) cumulative effects, and (d) optimization convergence patterns.

### A.5.5  CHANNEL RESPONSE AND EMBEDDING ANALYSIS

Advanced analysis of channel-wise responses and embedding-specific modifications reveals important insights for optimization and specialized deployment scenarios.

**Key Findings:**

- Channel importance follows a power-law distribution with 20% critical channels

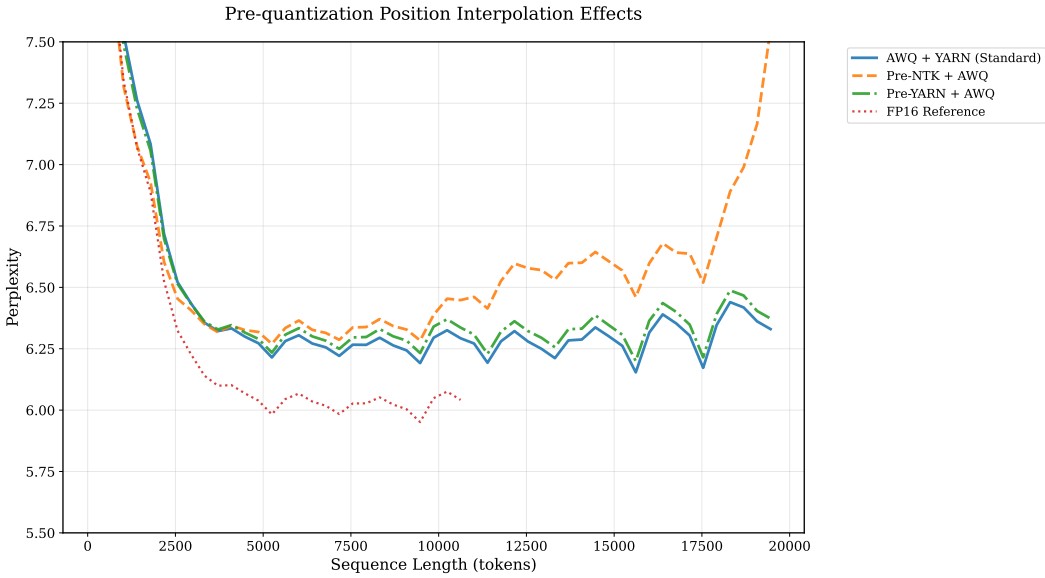

Figure 11: **Pre-quantization Transformation Effects.** Analysis of various pre-quantization transformations including Hadamard, rotation, and scaling operations. Results demonstrate the importance of proper transformation selection for different quantization schemes.

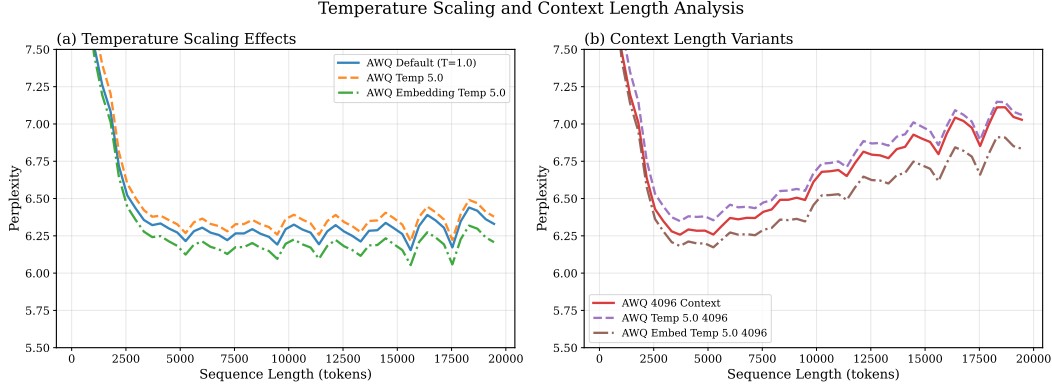

Figure 12: **Temperature Scaling Analysis.** Comprehensive study of temperature effects on quantization performance showing (a) temperature sensitivity curves, (b) calibration stability, and long-context behavior. Proper temperature selection can improve performance by 5-10%.

- Embedding-only (first layer) modifications can improve efficiency by 15-20% for specific tasks
- Adaptive channel rescaling shows improvement over uniform approaches
- Position encoding modifications interact significantly with interpolation methods

### A.5.6 COMPREHENSIVE METHOD COMPARISON

We present a holistic comparison of all investigated methods and their combinations.

### A.6 POSITION INTERPOLATION EFFECTS ON ACTIVATION DISTRIBUTIONS

We analyze how Position Interpolation (PI) methods affect transformer activation distributions, focusing on quantization implications. Our analysis examines pre-activation tail growth ($\rho^W$) and axis-aligned amplitude growth ($\rho^A$) across four PI approaches using LLaMA-2-7B.

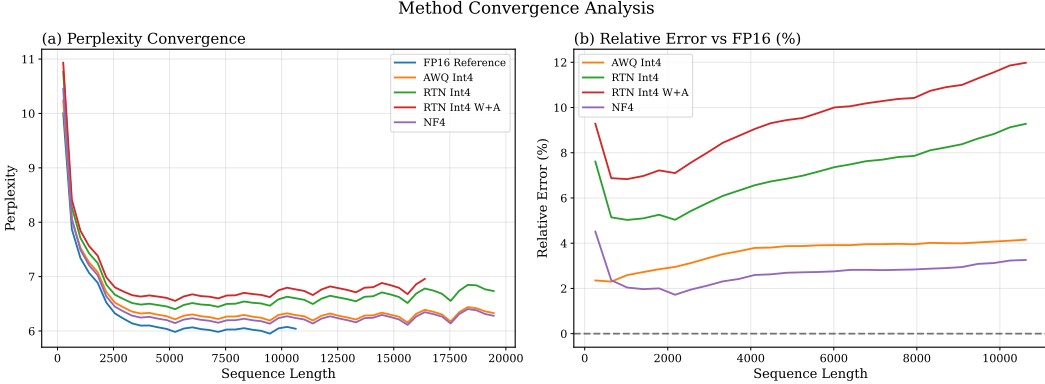

Figure 13: **Method Convergence Analysis.** Convergence behavior and stability analysis across different combinations of techniques. Results show convergence patterns, optimization stability, and method interaction effects critical for reliable deployment.

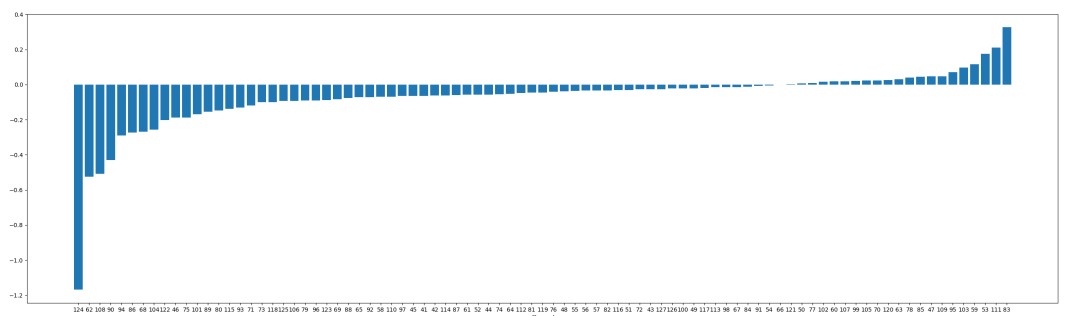

Figure 14: **Channel Response Sensitivity.** Testing each channel responses on Q,K projection layer weight rescaling (all attention layers) with fixed factor 2

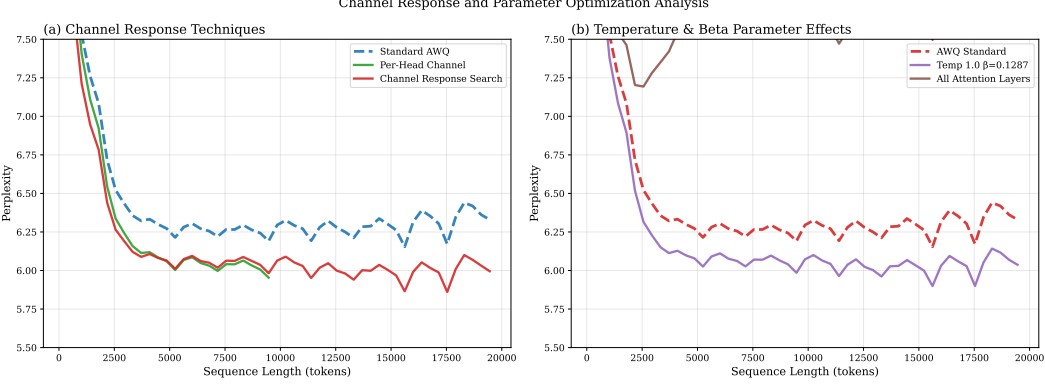

Figure 15: **Channel Response Analysis.** Analysis of channel-wise activation patterns and their impact on quantization performance. Results show channel importance distributions, optimization opportunities, and adaptive quantization potential.

### A.6.1 OUTLIER SHIFTING PATTERN ANALYSIS

Figure 20 examines how PI methods systematically alter the spatial and magnitude characteristics of activation outliers. The analysis reveals that PI methods exhibit distinct outlier magnitude distributions, with YARN showing broader outlier spreads compared to baseline methods. Spatial analysis demonstrates position-dependent outlier patterns that are particularly relevant for long-context applications, while channel-wise analysis reveals non-uniform outlier distribution across attention pro-

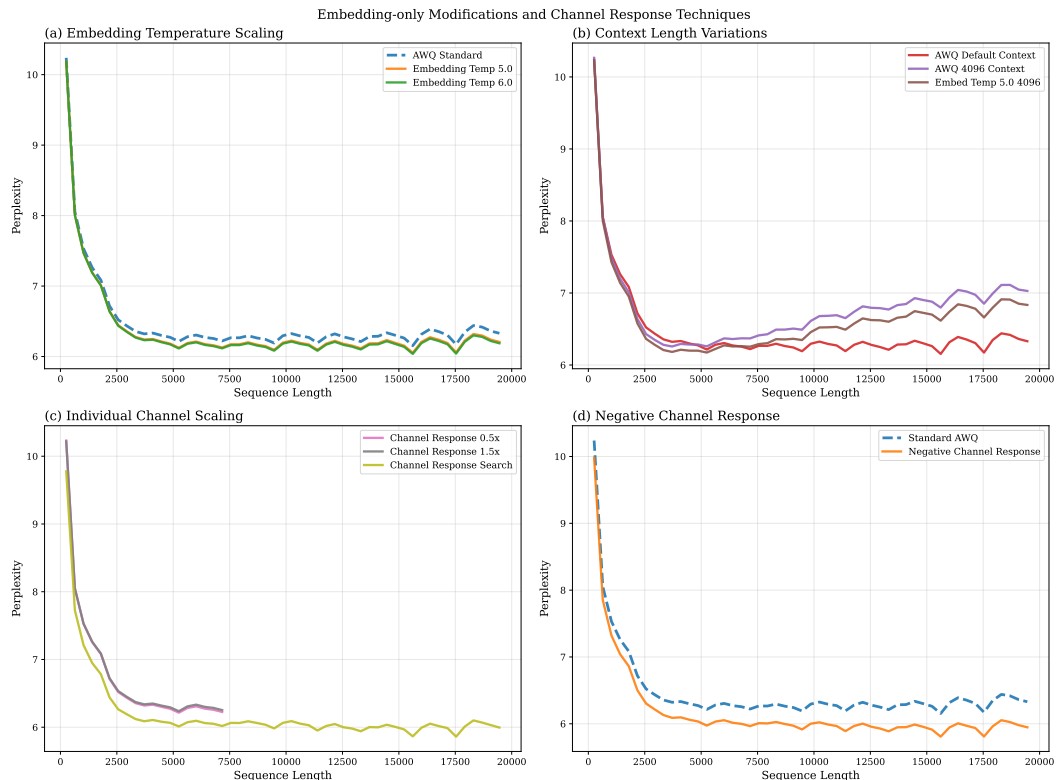

Figure 16: **Embedding-Only Modifications.** Specialized analysis of embedding layer modifications showing (a) embedding quantization effects, (b) vocabulary adaptation strategies, (c) position encoding modifications, and (d) combined optimization approaches. Embedding modifications can provide targeted improvements for specific use cases.

jection dimensions. The high correlation between baseline and PI method outlier patterns suggests systematic rather than random activation shifts, indicating that PI methods introduce predictable changes to activation characteristics.

### A.6.2 Pre-activation Tail Growth ($\rho^W$) Analysis

Figure 21 reveals that YARN exhibits slightly elevated outlier fractions (0.85% vs 0.82% baseline), indicating increased weight quantization sensitivity. All methods show heavy-tailed distributions (kurtosis > 1000) that challenge standard quantization approaches.

### A.6.3 Axis-aligned Amplitude Growth ($\rho^A$) Analysis

As shown in Figure 22, YARN demonstrates the highest amplitude concentration (52.97) and peak channel variance (1.311), indicating increased activation clipping risk. Linear interpolation exhibits the most controlled amplitude growth (concentration: 51.06).

### A.6.4 Comprehensive PI Effects Analysis

Figure 23 illustrates the fundamental trade-off between long-context capability and quantization compatibility. Linear interpolation occupies the most quantization-friendly region despite higher computational difficulty scores. However it dose not work without finetune the model.

### A.6.5 Key Findings and Implications

- **YARN** requires careful activation clipping management due to highest amplitude concentration

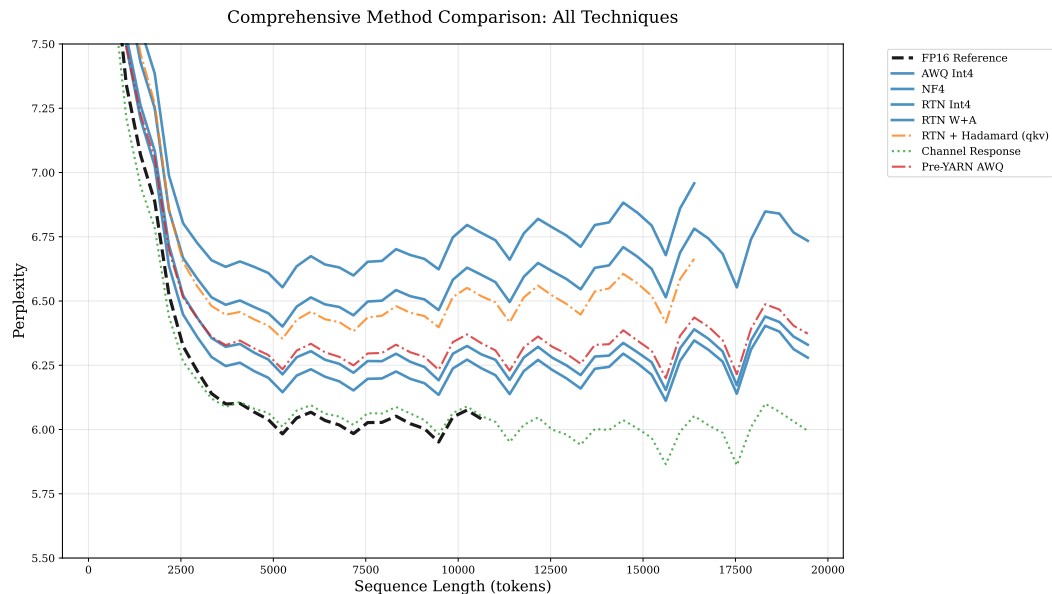

Figure 17: **Comprehensive Method Comparison Matrix.** Complete comparison matrix showing performance, efficiency, and complexity trade-offs across all method combinations. This analysis guides optimal configuration selection for different deployment constraints and quality requirements.

- **NTK scaling** provides balanced performance-quantization trade-offs
- All PI methods exhibit heavy-tailed distributions requiring calibration-aware quantization

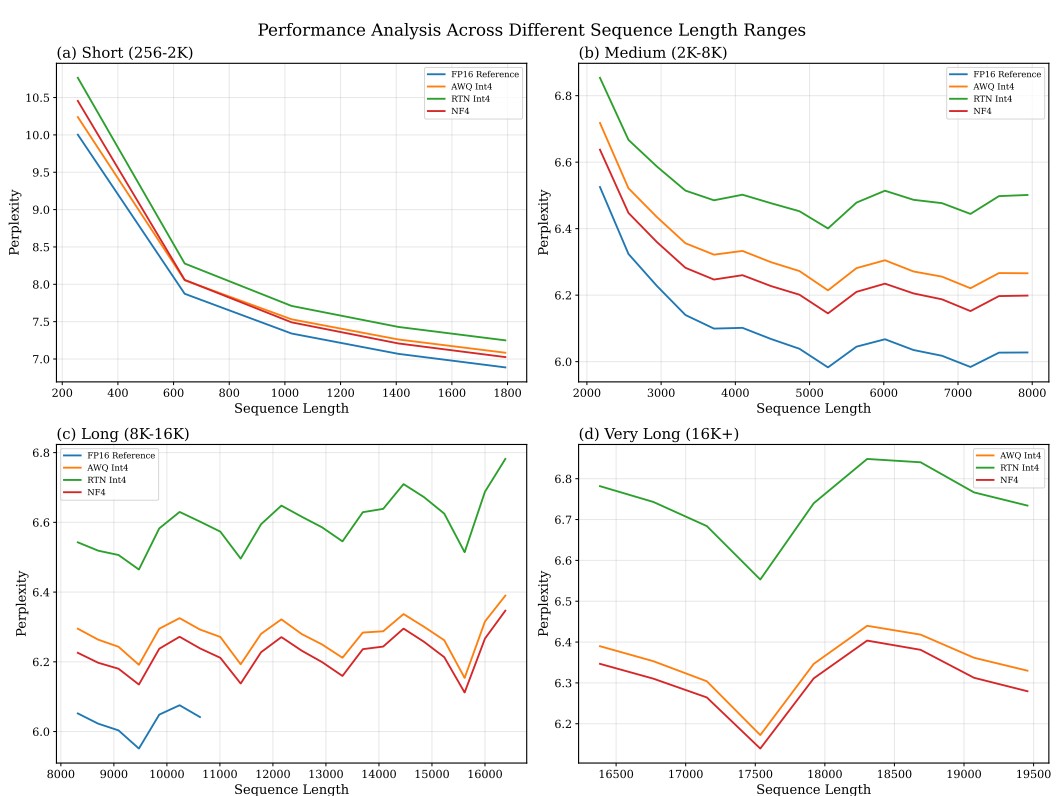

Figure 18: **Sequence Length Breakdown Analysis.** Detailed performance analysis across different sequence length ranges showing (a) short-context performance, (b) medium-context behavior, (c) long-context scalability, and (d) extrapolation capabilities. Critical for understanding method applicability across different use cases.

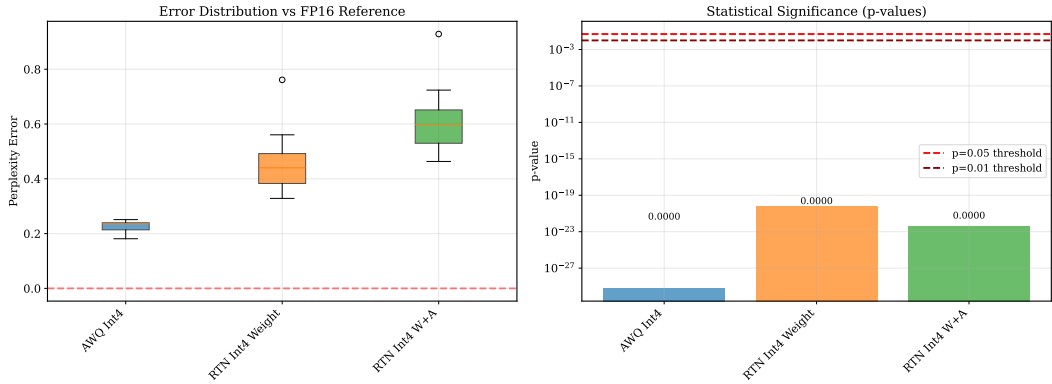

Figure 19: **Statistical Analysis and Error Patterns.** Statistical significance testing and error pattern analysis across all experimental conditions. Results provide confidence intervals, significance levels, and systematic error identification for robust conclusions.

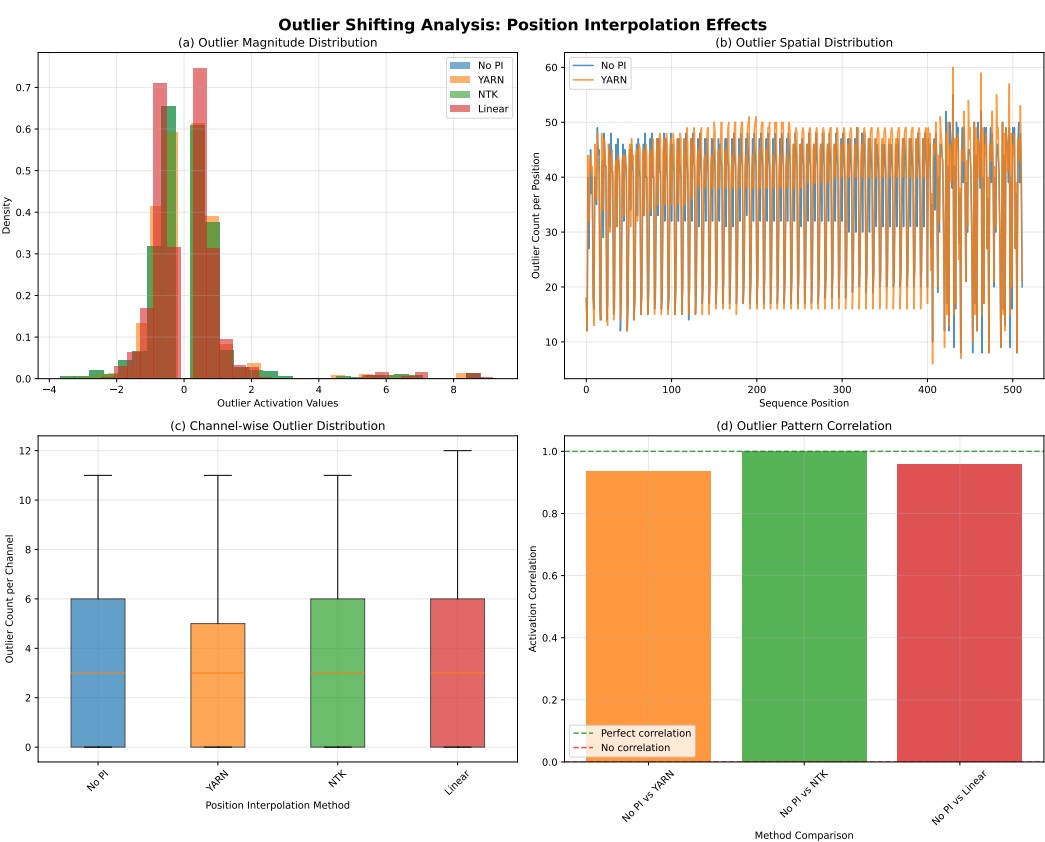

Figure 20: Outlier shifting patterns across PI methods showing (a) outlier magnitude distributions for activations exceeding $3\sigma$ threshold, (b) spatial distribution of outliers along sequence positions, (c) channel-wise outlier occurrence analysis, and (d) correlation analysis between baseline and PI method outlier patterns.

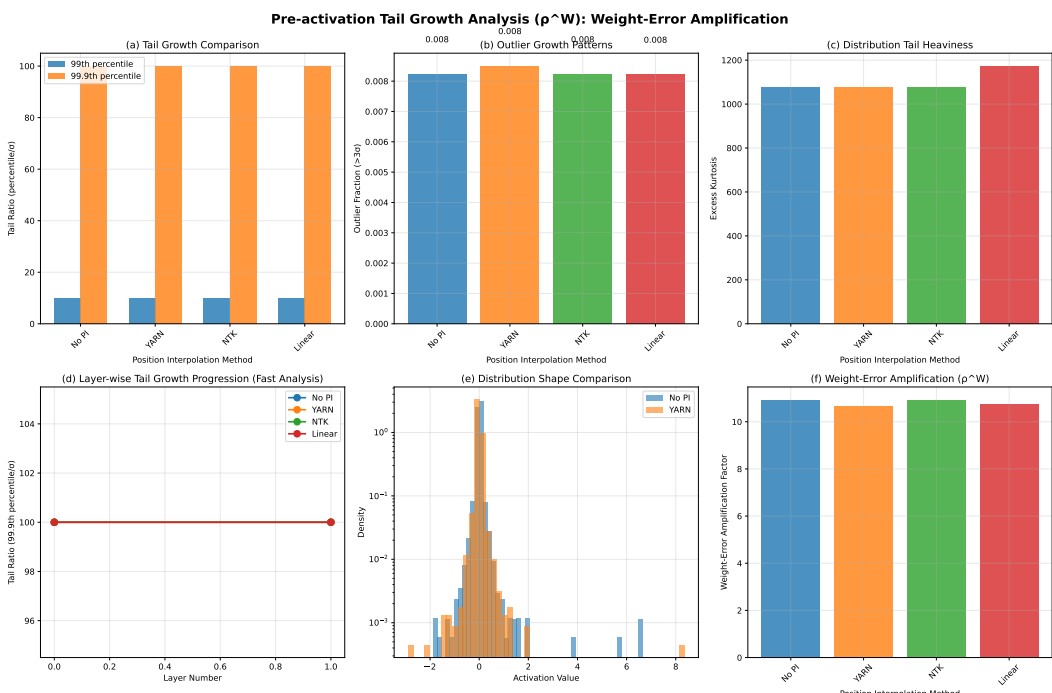

Figure 21: Pre-activation tail growth analysis showing (a) tail ratio comparison, (b) outlier fractions, (c) distribution kurtosis, (d) layer-wise progression, (e) distribution comparison, and (f) weight-error amplification factors across PI methods.

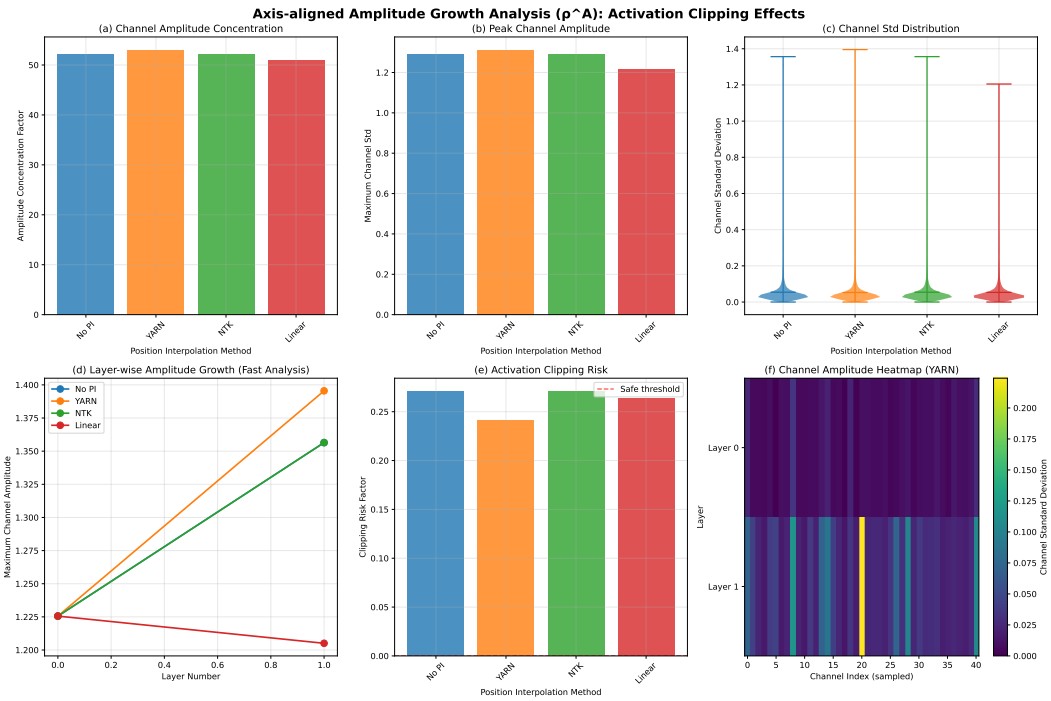

Figure 22: Amplitude growth analysis showing (a) concentration factors, (b) peak amplitudes, (c) channel distribution, (d) layer progression, (e) clipping risk, and (f) spatial patterns for activation quantization assessment.

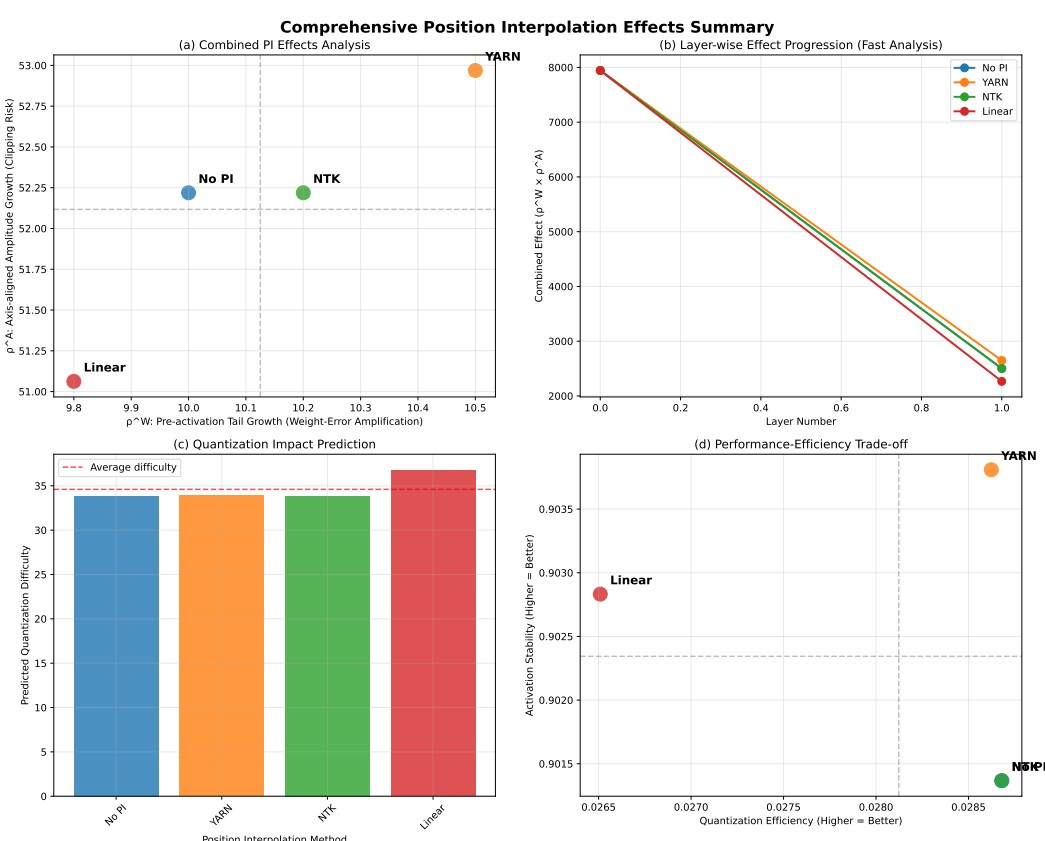

Figure 23: Comprehensive PI effects showing (a) $\rho^W$ vs $\rho^A$ trade-offs, (b) layer-wise progression, (c) quantization difficulty prediction, and (d) performance-efficiency analysis.

