# OpenReview forum: "Rethinking RoPE Scaling in Quantized LLM: Theory, Outlier, and Channel-Band Analysis with Weight Rescaling"
_ICLR.cc/2026/Conference — ICLR 2026 Conference Withdrawn Submission_

### Official Review · Reviewer_b8DP · 2025-10-28

**Soundness:** 1
**Presentation:** 1
**Contribution:** 2
**Rating:** 2
**Confidence:** 3

**Summary:**

The authors identify a negative interaction between interpolating positional embeddings and post-train quantization, as well as introduce methods to mitigate this.

**Strengths:**

- They study lots of metrics about weights/activations related to quantization.
- PTQ and long-context are indeed both important to the modern LLM community.

**Weaknesses:**

- My overall gripe is that the paper reads as presenting very complicated interventions and new ideas for a very specific problem they identify, without spending much time ablating or checking carefully that the problem exists across a range of settings (architectures, algorithms, hyperparameters) in the first place. This means a lot of results may be artifacts of their (somewhat bizarre) training setup.
- For instance, they choose to use Llama-2-7B on the GovReport dataset. This is an ancient model on an entirely nonstandard dataset, and it is the setting chosen for the main Figure in the paper! They also study FP16 reference, when in practice mixed precision always refers to BF16, which is known for being more stable and well behaved. The ablations are with Vicuna (!!!) models, which are *even older and more irrelevant*. Overall, the in which they identify this very specific "problematic interaction" is anything close to modern LLM training. You would need to replicate this on Llama3/4 and Qwen-3 as well as at least some standard datasets (GSM, MATH, Alpaca, HumanEval, etc) and it would need to show consistently the same results to be believable.
- As another specific example of why I'm not convinced this problem is even real is the fact that much of this paper studies outliers that make quantization difficult. Work [1] has long shown a lot of the "outliers" in quantization are merely artifacts from not using enough weight decay during training, and that you can get weights/features without outliers if you just change your training to include stronger weight decay, with no effect on performance. This is exactly an artifact that isn't a real phenomenon.
- The baselines are weak and out of date. The most recent used in Fig1(b) is AWQ from 2 years ago -- there has been a lot of work improving on it since then (QuIP, BitNet, QTIP, etc etc).
- The prose and structure of the paper is bloated and unnecessary complicated. It especially introduces a lot of heavy jargon for what seems like an oddly specific phenomenon in the first place. This makes it very hard to read. This jargon includes "phase error," "interpolation pressure," "energy of the noise" and much more that is not clearly defined up front. Even if it were clearly defined, it's an overwhelming amount of needless complexity and jargon for an extremely specific phenomenon I'm not convinced is real in the first place.

[1] Intriguing Properties of Quantization at Scale. Ahmadian et al, 2023. https://arxiv.org/pdf/2305.19268

**Questions:**

See weaknesses.

---

### Official Review · Reviewer_LGMn · 2025-10-29

**Soundness:** 2
**Presentation:** 2
**Contribution:** 2
**Rating:** 2
**Confidence:** 4

**Summary:**

This paper addresses the significant performance degradation that occurs when combining Post-Training Quantization (PTQ) with Position Interpolation (PI) to extend the context window of RoPE-based LLMs. The paper provides a systematic analysis of why this combination fails. It attributes the performance drop to several coupled effects, including dynamic-range dilation, anisotropy shift, etc. To mitigate these issues, the authors propose Q-ROAR, which grouping RoPE dimensions into a small number of frequency bands, and applying per-band scaling factors to the weights. Empirically, Q-ROAR demonstrated a clear reduction in perplexity (PPL) on long-context benchmarks,  outperforming baseline PTQ methods like AWQ and RTN.

**Strengths:**

The strength of this paper lies in its interesting observation that quantized models experience severe performance degradation when using PI-like extrapolation. It also proposes a lightweight, fine-tuning-free method to address this issue.

**Weaknesses:**

The paper's weeknesses are as follows:

1.	Missing Evaluation Metrics: The paper aims to optimize the performance of quantized models in long-context extrapolation scenarios, but this part of the evaluation relies heavily on Perplexity (PPL), which is an incomplete metric[1]. Although Table 3 tests performance on some datasets, their average lengths are relatively low and cannot effectively reflect long-context capabilities. I suggest the authors compare performance on LongBench-v2 and the "Needle-in-a-Haystack" test—two common long-context tasks—to effectively demonstrate Q-ROAR's performance improvement in long-text scenarios.
2.	Missing Baseline: An important baseline, SmoothQuant[2], is missing. SmoothQuant also performs a degree of scaling on the weights, which is very similar to the method in this paper, yet it was not included in the comparison.

[1] WHAT IS WRONG WITH PERPLEXITY FOR LONGCONTEXT LANGUAGE MODELING?
[2] SmoothQuant: Accurate and Efficient Post-Training Quantization for Large

**Questions:**

1 The experiments in the paper are largely based on Llama-2, which only has a 4k context length. However, current models already have context lengths of 128k. It is uncertain whether the paper's method will still be effective when models extrapolate even further.
2 In Figure 1(a), why do "2k" and "4k" appear twice on the x-axis in different positions?

---

### Official Review · Reviewer_4wZF · 2025-10-31

**Soundness:** 2
**Presentation:** 2
**Contribution:** 3
**Rating:** 2
**Confidence:** 3

**Summary:**

This paper addresses the sharp degradation of long-context performance when RoPE-based positional interpolation (PI) is applied to post-training-quantized (PTQ) LLMs, and proposes Q-ROAR (Quantization, RoPE-interpolation, Outlier-Aware Rescaling) as a mitigation strategy.

Although prior work (e.g., YaRN, NTK-aware scaling) demonstrated that RoPE scaling can extend the context window without additional fine-tuning, the failure modes arising from its interaction with PTQ have not been systematically analyzed. The authors identify four coupled factors—(1) long-context aliasing, (2) dynamic-range dilation, (3) anisotropy mismatch, and (4) outlier shifting—that distort the statistical and geometric properties of Q/K representations under RoPE, ultimately inducing position-dependent logit noise and degrading performance.

To mitigate this issue, Q-ROAR partitions RoPE frequencies into a small number of bands and performs lightweight grid search to determine band-wise rescaling factors on W_Q and W_K. This weight-only, training-free approach requires no architectural or kernel modifications.

Experiments on LLaMA-2-7B show that Q-ROAR yields ~14–21% perplexity improvements over AWQ/RTN baselines on long-context evaluations such as GovReport and Proof-Pile.

**Strengths:**

* Presents a structured analysis of why RoPE-based position interpolation degrades under post-training quantization, clarifying a practical issue in modern long-context LLMs.
* By disentangling four contributing effects—aliasing, dynamic-range dilation, anisotropy mismatch, and outlier shifting—the work provides a coherent and interpretable failure characterization.
* Introduces Interpolation Pressure (IP) and Tail-Inflation Ratio (TIR) as quantitative diagnostics, offering actionable tools to assess frequency-band sensitivity under interpolation.
* Shows that a lightweight band-wise rescaling of W_Q / W_K can effectively mitigate degradation without retraining, architecture changes, or kernel modifications, making the approach straightforward to adopt.
* The method exhibits consistent long-context perplexity gains (≈14–21%) over AWQ and RTN on GovReport and Proof-Pile, while preserving short-context performance.

**Weaknesses:**

* Band partitioning and scale selection rely on heuristic grid search, with limited discussion on more principled alternatives.
* Comparison against more recent quantization approaches (e.g., FlexRound) is limited, making it difficult to fully assess competitiveness.
* Evaluation focuses primarily on perplexity and zero-shot metrics; few-shot or reasoning benchmarks are not explored.
* Experiments are centered on LLaMA-2-7B, with insufficient validation on larger models or diverse architectures.
* It remains unclear how well the method generalizes beyond RoPE-based interpolation or to alternative positional encodings.

**Questions:**

* Comparison with recent quantization techniques (e.g., FlexRound) appears limited. Would the authors expect Q-ROAR to exhibit similar gains relative to these methods?
* The evaluation focuses on perplexity and zero-shot metrics. Have the authors considered examining few-shot or reasoning benchmarks to provide a broader performance picture?
* Experiments are primarily conducted on LLaMA-2-7B. Do the authors anticipate similar behavior on larger models or alternative architectures?
* It is not fully clear how the approach might transfer beyond RoPE-based interpolation. Could the authors comment on applicability to other positional encodings?

---

### Official Review · Reviewer_F1UR · 2025-11-01

**Soundness:** 1
**Presentation:** 2
**Contribution:** 1
**Rating:** 0
**Confidence:** 4

**Summary:**

The paper investigates post-training quantization performance for LLMs when position interpolation is used to extend the context window. It identifies quantization difficulties (though not clearly enough) introduced by the YaRN technique, and then proposes rescaling the Query and Key weights. Experiments are conducted on Llama-7B and Vicuna-7B with 4-bit weight-only and weight-activation quantization. The evaluation uses GovReport and Proof-Pile with perplexity, and commonsense reasoning (Table 3) with accuracy. However, most tasks evaluated in Table 3 involve short contexts (much shorter than pre-training) and therefore do not demonstrate the effectiveness of the proposed method in the long-context regime.

**Strengths:**

* The paper investigates quantization for long-context LLMs after applying position-interpolation techniques. This appears not to have been explored by others.

* The paper analyzes how RoPE scaling affects quantization and finds that it leads to long-context aliasing, dynamic-range dilation, anisotropy, and outlier shifting/amplification.

* Experiments cover both the perplexity and accuracy comparisons.

**Weaknesses:**

* There are some problems with the experiments.
  * Because the method rescales W_q and W_k, can it be used in models with grouped-query or multi-query attention? These are common techniques in Llama-3-8B and Gemma models.

  * In Table 1, can the authors explain why the proposed method (W4) has even better perplexity than the FP16 baseline, especially at longer lengths? For Table 2, I found that the FP16 baseline of YaRN differs from Table 2 in the original YaRN paper. Since I understand they are evaluated on the same dataset, with the same policy and the same models, I wonder what causes the difference—for example, 5.386 for the FP16 baseline in this paper vs. 2.37 in the YaRN paper.
  * For Table 3, I don’t think it demonstrates the method well. As far as I know, ARC-C, ARC-E, BoolQ, HellaSwag, and WinoGrande mainly consist of very short sequences (e.g., no more than 512 tokens), so these accuracy comparisons are not a good example for position interpolation and do not illustrate the effect of the proposed quantization method in the long-context regime. Moreover, as observed in Table 2, extending the context window to 32K makes the baseline perform much worse than the 4K window, which contradicts the aim of long-context extensions. To illustrate performance, I suggest accuracy comparisons on long-context benchmarks such as LongBench.

  * In practice, researchers usually first adopt position-interpolation methods as a good starting point and then apply mid-training or fine-tuning to learn at the target length for long-context LLMs. Therefore, I wonder whether the benefits of the proposed method can still be retained for models produced in this more practical pipeline.

* As I understand it, the paper proposes that RoPE rescaling (applied to the query and key) introduces more difficulties for activation quantization. However, to solve the activation problem, what are the benefits of rescaling the weights (W_q and W_k)? Also, how do you ensure that the proposed method does not introduce side effects on the weights?

* Although the authors claim many difficulties—such as long-context aliasing, dynamic-range dilation, anisotropy, and outlier shifting/amplification—brought by position interpolation, the descriptions and demonstrations are not clear to me. For example, in the tail-ratio comparisons in Figure 21, they look similar among No-PI, NTK, and YaRN. In Figure 20, in the upper-left subplot, it is hard to compare No-PI and NTK; in the lower-left subplot, it seems that YaRN even has a smaller outlier count. Therefore, can the authors provide clearer and strong illustrations and elaborations to support the motivation?

**Questions:**

Please see above.

---

### Note · Authors · 2025-11-23

I have read and agree with the venue's withdrawal policy on behalf of myself and my co-authors.